# FEDERATED LEARNING WITH BINARY NEURAL NETWORKS: COMPETITIVE ACCURACY AT A FRACTION OF THE COST

## ABSTRACT

Federated Learning (FL) preserves privacy by distributing training across devices. However, using DNNs is computationally demanding for the low-powered edge at inference. Edge deployment demands models that simultaneously optimize memory footprint and computational efficiency, a dilemma where conventional DNNs fail by exceeding resource limits. Traditional post-training binarization reduces model size but suffers from severe accuracy loss due to quantization errors. To address these challenges, we propose FedBNN, a rotation-aware binary neural network framework that learns binary representations directly during local training. By encoding each weight as a single bit $\{+1, -1\}$ instead of a 32-bit float, FedBNN shrinks the model footprint, significantly reducing runtime (during inference) FLOPs and memory requirements in comparison to federated methods using real models. Evaluations on multiple benchmark datasets demonstrate that FedBNN reduces resource consumption greatly while performing similarly to existing federated methods using real-valued models.

## 1 INTRODUCTION

Federated Learning (FL) has rapidly emerged as a cornerstone paradigm for privacy-preserving collaborative model training across distributed edge devices. In a standard FL workflow, a central server initializes a global model and communicates its parameters to participating clients. Each client, equipped with its own private dataset, trains the model locally before transmitting the updates back to the server, aggregating them to refine the global model iteratively. However, as modern deep learning models grow in scale and complexity, accommodating resource-constrained clients presents a fundamental challenge. Ensuring model efficiency is therefore critical during local training and especially for deployment on edge devices. Moreover, the frequent uplink and downlink communication inherent to FL often creates a severe bottleneck. Finally, the reliance on compact models in such settings further amplifies susceptibility to adversarial attacks, underscoring the importance of communication efficiency, lightweight design, and robustness in federated systems.

Several works focus on mitigating the communication overhead in federated learning. Li et al. (2025) enhances the efficiency of low-rank FL by addressing three critical challenges in decomposition by proposing Model Update Decomposition (MUD), Block-wise Kronecker Decomposition (BKD), and Aggregation-Aware Decomposition (AAD), which are complementary and can be jointly applied. Their approach demonstrates faster convergence with improved accuracy compared to prior low-rank baselines. Kim et al. (2024a) address unstable convergence under client heterogeneity and low participation by introducing a lookahead-gradient strategy. Their method broadcasts projected global updates without incurring extra communication costs or memory dependence, while additionally regularizing local updates to align with the overshot global model. This yields improved stability and tighter theoretical convergence guarantees, particularly under partial client participation. Hu et al. (2024) proposes a hybrid gradient compression (HGC) framework designed to reduce uplink and downlink costs by exploiting multiple forms of redundancy in the training process. With compression-ratio correction and dynamic momentum correction, HGC achieves a high compression ratio with negligible accuracy loss in practice.

Guo & Yang (2024) address generalization under client imbalance through Federated Group DRO algorithms to balance robustness and communication efficiency. Liu et al. (2024) propose FedLPA, a

one-shot aggregation framework that infers layer-wise Laplace posteriors to mitigate non-IID effects without requiring auxiliary data, markedly improving one-round training performance. Crawshaw & Liu (2024) studies more realistic client participation patterns and proposes Amplified SCAFFOLD, which achieves linear speedup and significantly fewer communication rounds via projected lookahead. Lu et al. (2025) introduces FedSMU, which simplifies communication by symbolizing updates (i.e., transmitting signs only) while decoupling the Lion optimizer between local and global steps, tackling communication and heterogeneity. Li et al. (2024) propose Federated Binarization-Aware Training (FedBAT), which directly learns binary model updates during local training through a stochastic, learnable operator $S(x, \alpha)$ with trainable step size $\alpha$. While this approach improves accuracy relative to post-training binarization, local optimization in FedBAT still relies on real-valued parameters, with binarization applied only to the communicated updates. Also, the final model learnt after training is real and more complex.

While communication efficiency is critical in federated learning (FL), maintaining lightweight models after training is equally important for resource-constrained edge devices. Kim et al. (2024b) proposes SpaFL, which introduces trainable per-filter thresholds to induce structured sparsity, requiring only threshold vectors to be uploaded. This leads to improved accuracy and reduced communication cost relative to sparse baselines. In contrast, our client-side BNNs employ binary filters ($\{-1, +1\}$), eliminating large computation and memory overhead. Lee & Jang (2025) develops BiPruneFL. This framework combines binary quantization with pruning to lower computation and communication costs, achieving up to two orders of magnitude efficiency gains while retaining accuracy comparable to uncompressed models. Similarly, Shah & Lau (2023) explores sparsification and quantization to address uplink and downlink communication, demonstrating superior trade-offs between model compression and accuracy preservation. Yang et al. (2021) specifically studies BNNs in FL, where clients transmit only binary parameters, and a Maximum Likelihood (ML) based reconstruction scheme is used to recover real-valued global parameters. Their framework effectively reduces communication costs while establishing theoretical convergence conditions for training federated BNNs.

In this work, we address the challenge of reducing runtime computational complexity in federated learning (FL) models on edge devices while maintaining high performance. Building on the idea of rotated binary neural networks Lin et al. (2020), we introduce FedBNN, a federated learning strategy inspired by FedAvg, which trains a rotated binary neural network with binary weights while preserving the same parameter count as its real-valued counterpart. Despite this parity, the binary representation of the global model yields substantial gains in memory efficiency and runtime computational savings.

Our main contributions are as follows:

1. We propose FedBNN, an FL framework for training Binary Neural Networks (BNNs) that achieve lower runtime computation and memory complexity compared to real-valued models.

2. We comprehensively compare FedBNN with state-of-the-art methods, conducting experiments on diverse benchmark datasets including FMNIST, SVHN, and CIFAR-10. We also consider data heterogeneity and perform comparisons with three types of data distribution.

3. We evaluate the runtime complexity of FedBNN in terms of computation cost and memory usage, demonstrating its efficiency advantages over existing approaches.

## 2 PRELIMINARIES

### 2.1 FEDERATED LEARNING

Federated Learning is a distributed machine learning paradigm that enables multiple clients to collaboratively train a shared model while keeping their data decentralized. Unlike traditional centralized learning, FL addresses critical challenges including data privacy, communication constraints, and statistical heterogeneity across participants. McMahan et al. (2017) introduced the Federated Averaging (FedAvg) algorithm, which combines local stochastic gradient descent on individual clients with periodic model averaging on a central server. The method addresses the fundamen-

tal optimization problem:

$$\min_{\mathbf{w} \in \mathbb{R}^d} \mathcal{L}(\mathbf{w}) \quad \text{where} \quad \mathcal{L}(\mathbf{w}) = \frac{1}{N_s} \sum_{i=1}^{N_s} \mathcal{L}_i(\mathbf{w}) \tag{1}$$

Here, $\mathcal{L}_i$ is the loss for a particular sample $(x_i, y_i)$, $\mathbf{w}$ is the model parameter, $N_s$ is the total number of samples. In the federated setting with $N_k$ clients, this is reformulated as:

$$\mathcal{L}(\mathbf{w}) = \sum_{k=1}^{N_k} \frac{N_{sk}}{N_s} l_k(\mathbf{w}) \quad \text{where,} \quad l_k(\mathbf{w}) = \frac{1}{N_{sk}} \sum_{i \in \mathcal{P}_k} \mathcal{L}_i(\mathbf{w}) \tag{2}$$

Here, $\mathcal{P}_k$ represents the data partition on client $k$ and $N_{sk} = |\mathcal{P}_k|$. The FedAvg algorithm operates by selecting a fraction $N_{cr}$ of clients each round, having each perform $N_e$ local epochs of SGD with batch size $N_b$:

$$\mathbf{w} \leftarrow \mathbf{w} - \eta \nabla \mathcal{L}(\mathbf{w}; b) \tag{3}$$

for each batch $b$, followed by server-side weighted averaging:

$$\mathbf{w}_{t+1} \leftarrow \sum_{k=1}^{N_k} \frac{N_{sk}}{N_s} \mathbf{w}_{t+1}^k \tag{4}$$

## 2.2 BINARIZED NEURAL NETWORK

If $g_\phi(\cdot)$ is a CNN with $L$ layers, its parameters are given by $\phi = \{\mathbf{W}_1, \ldots, \mathbf{W}_L\}$, where $\mathbf{W}_l \in \mathbb{R}^{c_o \times c_i \times k \times k}$ represents the weight matrix of the $l^{\text{th}}$ layer. Here, $c_i$ and $c_o$ represent the input and output channels, respectively, and $k$ denotes the filter size. In a Binary Neural Network (BNN), both weights ($\mathbf{W}_l$) and activations ($\mathbf{a}_l$) are binarized using the sign function:

$$\mathbf{W}_l^b = \text{sign}(\mathbf{W}_l), \quad \mathbf{a}_l^b = \text{sign}(\mathbf{a}_l), \tag{5}$$

and the convolution is approximated using bit-wise operations:

$$\mathbf{W}_l * \mathbf{a}_l \approx \mathbf{W}_l^b \circledast \mathbf{a}_l^b, \tag{6}$$

where $\circledast$ denotes bit-wise convolution (e.g., XNOR and bit count). Although the forward pass uses binarized values, real-valued weights and gradients are retained for backpropagation. Due to the non-differentiability of the sign function, whose derivative is zero almost everywhere, training binarized neural networks poses significant challenges, particularly in backpropagation, where meaningful gradients are required. Hence, a straight-through estimator (STE) is used: if $b = \text{sign}(r)$, then

$$\nabla_r = \nabla_b \cdot \mathbf{1}_{|r| \leq 1}, \tag{7}$$

where $\nabla_r = \frac{\partial C}{\partial r}$, $\nabla_b = \frac{\partial C}{\partial b}$, and $C$ is the cost function. To ensure stable updates, real weights are clipped to the range $[-1, 1]$. We adopt the approach from Hubara et al. (2016) to implement binarized convolution layers, converting floating-point operations into efficient XNOR and bit-count operations. While this drastically reduces computation and memory usage, it often comes at the cost of reduced accuracy. One key limitation of BNNs is the quantization error caused by binarizing the weight vector $\mathbf{w}_l \in \mathbb{R}^{n_l}$, which is the flattened form of $\mathbf{W}_l$, where $n_l = c_o \cdot c_i \cdot k^2$. This error arises due to the angular bias ($\phi$) between $\mathbf{w}_l$ and its binarized version $\mathbf{w}_l^b$, potentially degrading network performance.

# 3 PROPOSED METHOD - FEDBNN

## 3.1 ROTATED BINARY NEURAL NETWORK

### 3.1.1 TRAINABLE ROTATION WEIGHT WITH GLOBAL MEMORY

To address the angular bias, Lin et al. (2020) proposed applying a rotation matrix $\mathbf{R}_l \in \mathbb{R}^{n_l \times n_l}$ at the start of each training epoch to minimize the angle $\phi_l$ between $(\mathbf{R}_l)^T \mathbf{w}_l$ and $\text{sign}((\mathbf{R}_l)^T \mathbf{w}_l)$.

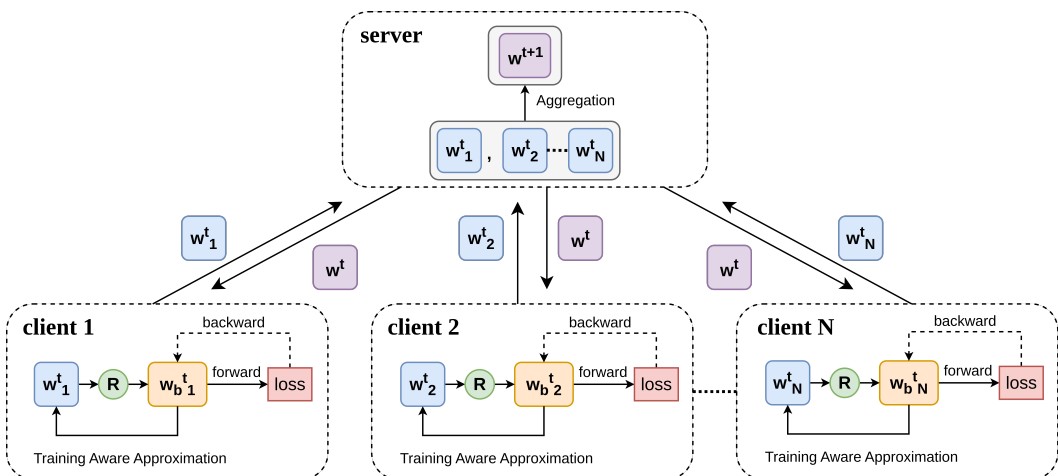

Figure 1: FedBNN overall architecture.

Building on this for our federated extension, instead of rotating the local client weight $\mathbf{w}_l$ directly, we first construct a fused weight $\mathbf{w} = \lambda_l \mathbf{w}_l + (1 - \lambda_l)\mathbf{w}_{\text{server}}$ using a trainable parameter $\lambda_l \in [0, 1]$ that interpolates between the client and server weight representations. The rotation is then applied to this fused vector $\mathbf{w}$, thus aligning the quantization with a federated-aware representation. This rotation is applied to the weights of each layer in every epoch of every round, as shown in Figure 1. For simplicity, we omit subscripts denoting the layer, client, or epoch in the following discussion. To minimize the angular bias ($\phi$), $\cos(\phi)$ needs to be maximized and formulated as follows:

$$\cos\left(\phi\right) = \frac{\text{sign}((\mathbf{R})^T \mathbf{w})^T ((\mathbf{R})^T \mathbf{w})}{\|\text{sign}((\mathbf{R})^T \mathbf{w})\|_2 \|((\mathbf{R})^T \mathbf{w})\|_2}, \tag{8}$$

where $(\mathbf{R})^T \mathbf{R} = \mathbf{I}_n$ is the $n$-th order identity matrix. Note, $\|\text{sign}((\mathbf{R})^T\mathbf{w})\|_2 = \sqrt{n}$ and $\|((\mathbf{R})^T\mathbf{w})\|_2 = \|\mathbf{w}\|_2$. Since the training happens at the beginning of each epoch, we can take $\|\mathbf{w}\|_2$ to be a constant. With the help of algebraic manipulations, we get $\mathbf{W}'^b = \text{sign}((\mathbf{R}_1)^T \overline{\mathbf{W}} \mathbf{R}_2)$, $\text{Vec}(\overline{\mathbf{W}}) = \mathbf{w}$, $\overline{\mathbf{W}} \in \mathbb{R}^{n_1 \times n_2}$ and

$$\mathbf{w}^T \mathbf{R} = \mathbf{w}^T (\mathbf{R}_1 \otimes \mathbf{R}_2) = \text{Vec}((\mathbf{R}_2)^T (\overline{\mathbf{W}})^T \mathbf{R}_1)$$

where $\otimes$ is the Kronecker product and the operation $\text{Vec}(\cdot)$ vectorizes an input matrix. The final optimization objective is given by

$$\begin{aligned} \underset{\mathbf{W}'^b, \mathbf{R}_1, \mathbf{R}_2}{\arg\max} \ \ & \text{tr}(\mathbf{W}'^b (\mathbf{R}_2)^T (\overline{\mathbf{W}})^T \mathbf{R}_1) \\ \text{s.t. } & \mathbf{W}'^b \in \{+1, -1\}^{n_1 \times n_2} \\ & (\mathbf{R}_1)^T \mathbf{R}_1 = \mathbf{I}_{n_1} \\ & (\mathbf{R}_2)^T \mathbf{R}_2 = \mathbf{I}_{n_2}. \end{aligned} \tag{9}$$

Since the above optimization is a non convex problem, an alternating optimization approach is used, where one variable is updated, keeping the other two fixed until convergence. We, therefore, have three steps in each, as shown in Algorithm 1:

1. The first step is to learn $\mathbf{W}'^b$ while fixing $\mathbf{R}_1$ and $\mathbf{R}_2$. It is solved by

$$\mathbf{W}'^b = \text{sign}((\mathbf{R}_1)^T \overline{\mathbf{W}} \mathbf{R}_2) \tag{10}$$

2. The next step updates $\mathbf{R}_1$ while keeping $\mathbf{W}'^b$ and $\mathbf{R}_2$ constant. Performing $\text{SVD}\left(\mathbf{W}'^b (\mathbf{R}_2)^T (\overline{\mathbf{W}})^T\right) = \mathbf{U}_1 \mathbf{S}_1 (\mathbf{V}_1)^T$, it is solved by

---

**Algorithm 1:** Federated Binary Neural Network (FedBNN) training. The $N_k$ clients are indexed by $k$; $N_b$ is the local minibatch size, $N_e$ is the number of local epochs, and $\eta$ is the learning rate. $N_{cr}$ is the number of clients selected per round. $N_{eR}$ is the number of epochs of rotation. $N_{lR}$ is the number of layers that require rotation. $\Theta$ is the set of all trainable parameters.

---

**Server executes:**
initialize $\mathbf{w}_0$
**for** *round* **in** *range*($N_r$) **do**
   $S_t \leftarrow$ (random set of $N_{cr}$ clients)
   **for** *each client* $\in S_t$ **in parallel do**
      $\mathbf{w}_{t+1}^k \leftarrow$ **ClientUpdate** $(k, \mathbf{w}_t)$
   **end**
   $\mathbf{w}_{t+1} = \sum_{k=1}^K \frac{N_{sk}}{N_k} \mathbf{w}_{t+1}^k$
**end**

**ClientUpdate** $(k, \mathbf{w}_{\text{server}})$:
$\mathcal{B} \leftarrow$ (split $\mathcal{P}_k$ into batches of size $N_b$)
$\mathbf{w} \leftarrow \mathbf{w}_{\text{server}}$
**for** *epoch* **in** *range*($N_e$) **do**
   **for** $e_R$ **in** *range*($N_{eR}$) **do**
      **for** $l$ **in** *range*($N_{lR}$) **do**
         $\mathbf{W}_l'^b \leftarrow \text{sign}((\mathbf{R}_{l1})^T \overline{\mathbf{W}}_l \mathbf{R}_{l2})$
         $\mathbf{U}_1, \mathbf{S}_1, \mathbf{V}_1 \leftarrow \text{SVD}(\mathbf{W}_l'^b (\mathbf{R}_{l2})^T (\overline{\mathbf{W}}_l)^T)$
         $\mathbf{R}_{l1} \leftarrow \mathbf{V}_1 (\mathbf{U}_1)^T$
         $\mathbf{U}_2, \mathbf{S}_2, \mathbf{V}_2 \leftarrow \text{SVD}((\overline{\mathbf{W}}_l)^T \mathbf{R}_{l1} \mathbf{W}_l'^b)$
         $\mathbf{R}_{l2} \leftarrow \mathbf{U}_2 (\mathbf{V}_2)^T$
      **end**
   **end**
   **for** *batch* $b \in \mathcal{B}$ **do**
      **for** $\theta \in \Theta$ **do**
         $\theta \leftarrow \theta - \eta \, \sigma_\theta(b)$
      **end**
   **end**
**end**
return $\mathbf{w}$ to server

---

$$\mathbf{R}_1 = \mathbf{V}_1 (\mathbf{U}_1)^T. \tag{11}$$

3. Similar to the previous steps, the following step updates $\mathbf{R}_2$ while keeping $\mathbf{W}'^b$ and $\mathbf{R}_1$ constant. Performing $\text{SVD}\left((\overline{\mathbf{W}})^T \mathbf{R}_1 \mathbf{W}'^b\right) = \mathbf{U}_2 \mathbf{S}_2 (\mathbf{V}_2)^T$, it is solved by

$$\mathbf{R}_2 = \mathbf{U}_2 (\mathbf{V}_2)^T \tag{12}$$

### 3.1.2 ADJUSTABLE ROTATED WEIGHT VECTOR WITH GLOBAL MEMORY

The optimization steps described above are executed iteratively. As noted in Lin et al. (2020), the variables $\mathbf{W}'^b$, $\mathbf{R}_1$, and $\mathbf{R}_2$ typically converge within three iterations. However, the process may still get trapped in a local optimum due to overshooting/undershooting. To mitigate this, Lin et al. (2020) introduced an adjustable rotated weight vector scheme to further reduce angular bias after the bi-rotation step. However, in a federated setting, a client may need to align its weights not just with its own rotated representation but also with $\mathbf{w}_{\text{server}}$. To this end, we propose a generalized update:

$$\tilde{\mathbf{w}} = \mathbf{w} + \alpha(\mathbf{R}^T \mathbf{w} - \mathbf{w}) + \beta(\mathbf{w}_{\text{server}} - \mathbf{w}) \tag{13}$$

where $\mathbf{w}$ is the interpolated weight, $\alpha = \left|\sin(\theta)\right|$, $\beta = \left|\sin(\gamma)\right|$, $\theta$, $\gamma \in \mathbb{R}$ and $\alpha$, $\beta \in [0, 1]$. Here, $\alpha$ and $\beta$ are learnable parameters controlling the contributions from the rotated and server directions,

respectively. The added regularization term ($\beta(\mathbf{w}_{\text{server}} - \mathbf{w})$) updates adaptively and fuses local and global knowledge while correcting angular bias with respect to both. It gathers inspiration from Li et al. (2020), where a proximal term is added to prevent training divergence due to heterogeneous data. While the original method in Lin et al. (2020) is designed for centralized training, we extend this framework to federated learning.

In summary, each client receives the global server weight $\mathbf{w}_{\text{server}}$ at the beginning of each round. We introduce a learnable fusion parameter $\lambda_l$ to interpolate between the client and server weights, forming a federated-aware fused weight $\mathbf{w}$. The bi-rotation is then applied to $\mathbf{w}$ instead of $\mathbf{w}_l$, allowing angular correction in the shared representation space. Moreover, we introduce two additional learnable scalars $\alpha_l$ and $\beta_l$ to adaptively adjust the influence of the rotated direction and the global server model, respectively, during the binarization step.

### 3.1.3 TRAINING AWARE APPROXIMATION FOR FEDERATED LEARNING

To improve upon the STE, Lin et al. (2020) introduced a training-aware approximation function that serves as a smooth, epoch-dependent surrogate for the sign function, enabling better gradient flow during early training. Unlike centralized training, where $t$ and $k$ are updated locally each epoch, our federated setup maintains these values across global rounds to ensure consistent client training behavior. The approximation is given by:

$$F(x) = \begin{cases} k \cdot \left( -\text{sign}(x) \cdot \dfrac{t^2 x^2}{2} + \sqrt{2}tx \right), & \text{if } |x| < \dfrac{\sqrt{2}}{t}, \\ k \cdot \text{sign}(x), & \text{otherwise}, \end{cases} \tag{14}$$

where the coefficients $t$ and $k$ evolve with training as:

$$t = 10^{(T_{\min}) + \frac{(rN_e + e)}{N_r N_e}(T_{\max} - T_{\min})} \tag{15}$$

$$k = \max\left( \frac{1}{t}, 1 \right) \tag{16}$$

where $T_{\min} = -2$, $T_{\max} = 1$, $N_r$ the total number of global training rounds, and $r$ the current round index, $N_e$ the total number of local training epochs, and $e$ the current epoch of training. The derivative of this function with respect to $x$ is:

$$F'(x) = \frac{\partial F(x)}{\partial x} = \max\left( k \cdot (\sqrt{2}t - |t^2 x|), 0 \right), \tag{17}$$

which yields non-zero gradients during early training, allowing effective optimization of both client and server-side parameters, and progressively transitions to a sign-like function, thus preserving binarization.

Using this surrogate, we compute gradients of the loss $\mathcal{L}$ with respect to both activations $\mathbf{a}$ and the mixed weights $\tilde{\mathbf{w}}$ as follows:

$$\sigma_{\mathbf{a}} = \frac{\partial \mathcal{L}}{\partial F(\mathbf{a})} \cdot \frac{\partial F(\mathbf{a})}{\mathbf{a}}, \tag{18}$$

$$\sigma_{\mathbf{w}} = \frac{\partial \mathcal{L}}{\partial F(\tilde{\mathbf{w}})} \cdot \frac{\partial F(\tilde{\mathbf{w}})}{\partial \tilde{\mathbf{w}}} \cdot \frac{\partial \tilde{\mathbf{w}}}{\partial \mathbf{w}}, \tag{19}$$

where the mixed-weight Jacobian is defined as:

$$\frac{\partial \tilde{\mathbf{w}}}{\partial \mathbf{w}} = (1 - \alpha - \beta) \cdot \mathbf{I}_n + \alpha \cdot \mathbf{R}^\top, \tag{20}$$

accounting for both the direct client path and the rotation-aligned correction. The gradients of the adaptive mixing parameters $\alpha$ and $\beta$, which respectively control the contributions from the rotation-aligned direction and the global server model, are computed as:

$$\sigma_\alpha = \frac{\partial \mathcal{L}}{\partial \tilde{\mathbf{w}}} \cdot (\mathbf{R}^\top \mathbf{w} - \mathbf{w}), \tag{21}$$

$$\sigma_\beta = \frac{\partial \mathcal{L}}{\partial \tilde{\mathbf{w}}} \cdot (\mathbf{w}_{\text{server}} - \mathbf{w}). \tag{22}$$

This training-aware formulation plays a critical role in stabilizing federated optimization by aligning binarization with geometric orientation and enabling meaningful gradient flow throughout local client training, as shown in Figure 1. In summary, at the beginning of every training epoch of each client, the rotation matrices, $\mathbf{R}_1$ and $\mathbf{R}_2$, are learned for a fixed $\mathbf{w}$. At the training phase, with the fixed $\mathbf{R}_1$ and $\mathbf{R}_2$, the NN takes the sign of parameter $\tilde{\mathbf{w}}$ for the forward pass and the parameters $\mathbf{w}_l$, $\alpha$ and $\beta$ are updated during back-propagation. Since $\alpha, \beta$ are also trainable parameters, it enables the network to learn a suitable value that further optimizes the application of the rotation in equation (16).

### 3.2 Aggregation of Rotated BNNs at the server

At the end of each training round, clients send their locally learned model weights to the server for aggregation using the FedAvg method. In addition to the layer weights, each client also transmits the Rotation matrices of its layers. Although this increases communication overhead, incorporating the aggregated rotation matrices on the client side in the next round leads to significant performance gains. While the averaged Rotation matrix is no longer orthogonal, it is corrected during the subsequent three-step rotation optimization. Importantly, the aggregated matrix captures information from all clients, helping to realign the weight vectors and adjust their norms for the next round. Other variants of the algorithm considered in our study are: 1) enforcing orthogonality of the aggregated Rotation matrix at the start of each round, before client-side rotation optimization, and 2) performing rotation optimization directly on the server, avoiding the transmission overhead of Rotation matrices after local training. The outcomes of this ablation analysis are summarized in Table 2. In the previous sections, we described the techniques applied on the client side within the proposed FedBNN framework. As outlined in Algorithm 1, rotation optimization is carried out at the start of each epoch to reduce quantization error before binarization. Additionally, constraining the deviation of the locally learned weight vector from the global model weights ($\mathbf{w}_{\text{server}}$) substantially enhances performance. Furthermore, the aggregated rotation matrix distributed by the server provides an effective initialization for the rotation optimization at the beginning of each round.

### 3.3 Runtime computation savings for binary models

From Shankar et al. (2024), we estimate the number of runtime multiplication and addition operations in a 2D CNN for comparison. For a convolution between real-valued $\mathbf{W}_l \in \mathbb{R}^{c_o \times c_i \times k \times k}$ and input $\mathbf{a}_l \in \mathbb{R}^{c_i \times h_{in}^w \times h_{in}^h}$, the output is $\mathbf{a}_{l+1} \in \mathbb{R}^{c_o \times h_{out}^w \times h_{out}^h}$. The number of multiplications is $c_i \cdot k^2 \cdot h_{out}^w \cdot h_{out}^h \cdot c_o$, and additions are roughly of the same order. Thus, the total FLOPs for the $l^{\text{th}}$ layer is approximately $2 \cdot c_i \cdot k^2 \cdot h_{out}^w \cdot h_{out}^h \cdot c_o$. Also, we consider every parameter to be of 32 bits. Hence, to calculate the total memory, we multiply the total parameters by 32. By binarizing weights and activations to $\{+1, -1\}$, convolutions are replaced by efficient XNOR and bit-count operations. And all the weights will only need 1 bit for storage. Rastegari et al. (2016) states that using binary neural networks results in a FLOPs reduction of $58\times$ and memory savings of $32\times$. Hence, to compare FedBNN with real models, we use this conversion factor in FLOPs and memory to estimate the computation savings.

## 4 Experimental Evaluation

### 4.1 Setup

All experiments are implemented using the PyTorch framework and executed on an NVIDIA A100 GPU. We conduct federated training with $N_c = 100$ clients participating in each experiment. The training follows the standard federated averaging (FedAvg) protocol for model synchronization. We use stochastic gradient descent with a learning rate of $0.1$ for optimization. The Learning rate is decreased by a multiplicative factor of 2 from round 200 every 50 rounds. Each federated training round comprises local training on selected clients for 10 epochs (5 epochs for FMNIST) with a mini-batch size of 64. 10 clients are randomly sampled for local model updates in each round. The global training process runs for a total of 500 rounds.

| Method | Dataset (Model) | Accuracy | | | FLOPs | Memory (MB) | Binarized Accuracy | | |
|---|---|---|---|---|---|---|---|---|---|
| | | IID | Non-IID 1 | Non-IID 2 | | | IID | Non-IID 1 | Non-IID 2 |
| FedAvg | | **92.24** | **91.44** | **89.28** | $2.02 \times 10^7$ | 1.5635 | 53.42 | 63.68 | 54.72 |
| FedBAT | FMNIST | 89.12 | 87.66 | 85.56 | $2.02 \times 10^7$ | 1.5635 | 14.34 | 16.98 | 8.0 |
| FedMud | (CNN4) | 89.60 | 88.60 | 86.00 | $2.02 \times 10^7$ | 1.6291 | 63.2 | 66.5 | 66.08 |
| FedBNN | | 88.24 | 85.80 | 82.10 | $\mathbf{3.48 \times 10^5}$ | 0.0489 | **73.42** | **80.58** | **67.8** |
| FedAvg | | **92.10** | **90.60** | **89.34** | $3.00 \times 10^7$ | 1.5965 | 28.01 | 22.56 | 16.92 |
| FedBAT | SVHN | 86.01 | 80.83 | 75.78 | $3.00 \times 10^7$ | 1.5965 | 50.35 | 26.69 | 34.71 |
| FedMud | (CNN4) | 86.31 | 84.38 | 83.14 | $3.00 \times 10^7$ | 1.6127 | 69.92 | 50.87 | 51.19 |
| FedBNN | | 85.40 | 84.42 | 81.93 | $\mathbf{5.19 \times 10^5}$ | 0.0498 | **84.09** | **81.94** | **79.88** |
| FedAvg | | **90.86** | **86.28** | **70.62** | $4.40 \times 10^8$ | 19.6170 | 17.2 | 11.38 | 12.74 |
| FedBAT | CIFAR10 | 89.38 | 72.80 | 63.70 | $4.40 \times 10^8$ | 19.6170 | 13.62 | 10.94 | 10.26 |
| FedMud | (ResNet-10) | 88.74 | 84.22 | 67.22 | $4.40 \times 10^8$ | 19.6170 | 15.54 | 10.78 | 18.98 |
| FedBNN | | 86.26 | 76.30 | 67.82 | $\mathbf{1.11 \times 10^7}$ | 0.6130 | **84.54** | **70.16** | **61.58** |

Table 1: Performance comparison for $N_c = 100$. The FLOPs and memory values are calculated during runtime. Binarized accuracy refers to the model's performance after the weights and activations have been binarized.

## 4.2 DATASETS AND PARTITIONING

To comprehensively evaluate the effectiveness of FedBNN, we conduct experiments on three widely used federated learning benchmarks, namely, FMNIST, SVHN, and CIFAR10. The client models are trained on the partitioned training data for all experiments. The testing data is split into two equal sets: validation and testing. The best model is picked at the server after aggregation based on the validation set. The final performance of the model is reported on the unseen test set. To thoroughly assess our approach's data heterogeneity performance, we evaluate under both IID and non-IID data distribution scenarios, following federated learning benchmarks McMahan et al. (2017). Under IID partitioning, each client is assigned an equal quantity of randomly sampled data, resulting in statistically similar local datasets. The non-IID setting comprises two configurations: Non-IID 1 and Non-IID 2. In Non-IID 1, samples are distributed among clients according to a Dirichlet distribution Hsu et al. (2019), with the Dirichlet parameter $\alpha$ modulating the degree of statistical skew, set to 0.3 for all the datasets. Non-IID 2 represents an extreme heterogeneity case, where each client receives data from only a subset of possible labels, specifically, 10 random labels per client for CIFAR-100 and 3 random labels per client for the other datasets. These partitioning strategies enable a systematic examination of model performance as data distributions on clients become increasingly disparate, closely mirroring realistic federated deployment scenarios.

## 4.3 SIMULATION RESULTS

To showcase the performance of FedBNN, we employ a CNN with four binarized convolution layers, one fully connected layer for FMNIST and SVHN, and a ResNet10 architecture for CIFAR10. Table 1 presents the classification accuracy across FMNIST, SVHN, and CIFAR10 datasets under IID and Non-IID data splits. FedAvg, having no binarization bottleneck in training or communication, consistently achieves the highest accuracy, with values such as 92.24% (IID, FMNIST) and 92.10% (IID, SVHN). FedBNN, although slightly lower, remains competitive within 10% of all real-valued methods. For example, on FMNIST under Non-IID 2, FedBNN attains 82.10% compared to 89.28% of FedAvg, a gap of only 7.18%. On SVHN IID data, FedBNN reaches 85.40% versus 92.10% for FedAvg, a difference of 6.7%, while under Non-IID 1 it achieves 84.42% against 90.60% (gap of 6.18%), and under Non-IID 2 81.93% compared to 89.34% (gap of 7.41%). On CIFAR10 IID data, FedBNN reaches 86.26% versus 90.86% for FedAvg, a difference of 4.6%. Notably, on CIFAR10 with Non-IID 1 data, FedBNN achieves 76.30%, which is 3.5% higher than FedBAT (72.80%). Also, FedBNN outperforms FedBAT by 4.12% for the CIFAR10 dataset NON-IID 2 distribution. The proposed method can even outperform certain baselines under challenging data distributions. These results demonstrate that FedBNN preserves reasonable accuracy despite aggressive compression and binarization.

| Method (Ablation) | Dataset (Model) | Clean Accuracy | | |
|---|---|---|---|---|
| | | IID | Non-IID 1 | Non-IID 2 |
| FedBNN (with orthogonal R1 R2 at client) | FMNIST (CNN4) | 83.64 | 84.90 | 77.60 |
| FedBNN (with server R1 R2 computation) | | 85.28 | 82.02 | 76.30 |
| FedBNN | | **88.24** | **85.80** | **82.10** |
| FedBNN (with orthogonal R1 R2 at client) | SVHN (CNN4) | 82.01 | 81.05 | 79.52 |
| FedBNN (with server R1 R2 computation) | | 76.28 | 74.32 | 72.46 |
| FedBNN | | **85.40** | **84.42** | **81.93** |
| FedBNN (with orthogonal R1 R2 at client) | CIFAR10 (ResNet10) | 85.64 | 74.30 | 65.40 |
| FedBNN (with server R1 R2 computation) | | 85.70 | 68.34 | 65.78 |
| FedBNN | | **86.26** | **76.30** | **67.82** |

Table 2: Ablation Study considering different rotation matrix initializations.

A significant advantage of FedBNN is the drastic reduction in runtime computational and memory requirements. FLOPs are reduced by nearly two orders of magnitude: for example, in FMNIST, FedBNN requires only $3.48 \times 10^5$ operations compared to $2.02 \times 10^7$ for FedAvg, a $\sim 58\times$ reduction. Similarly, in CIFAR10, FedBNN reduces FLOPs from $4.40 \times 10^8$ to $1.11 \times 10^7$, offering a $\sim 40\times$ improvement. Memory usage follows a similar trend, with FedBNN requiring only 0.0489 MB for FMNIST compared to 1.5635 MB in FedAvg, i.e., saving $32\times$. Even for the larger ResNet-10 model on CIFAR10, memory is reduced from 19.6 to 0.613 MB, yielding $32\times$ compression. These savings are particularly impactful for resource-constrained federated clients. In the next section, we will compare the performance of methods after post-training binarization.

Post-training binarization of real models will also lead to a binary model at the expense of performance. Since FedBNN incorporates binarization into training, despite the strong compression, FedBNN achieves superior binarized accuracy compared to other baselines. On FMNIST, FedBNN records 73.42% under IID, outperforming FedAvg (53.42%) and FedBAT (14.34%) by 20% and 59.08% respectively. For SVHN, FedBNN achieves 84.09% (IID), significantly higher than the 28.01% of FedAvg. Similarly, on CIFAR10, FedBNN maintains 84.54% binarized accuracy under IID, surpassing all baselines by a wide margin. Even in Non-IID 2 settings, FedBNN reaches 67.8% (FMNIST), 79.88% (SVHN), and 61.58% (CIFAR10), remaining much closer to the full-precision performance. These results highlight that FedBNN preserves competitive accuracy while drastically lowering computation and memory requirements, making it well-suited for federated learning with limited client resources.

Table 2 reports the results of two FedBNN variants against the standard formulation as discussed in Section 3.2. On FMNIST, the baseline FedBNN achieves 88.24% (IID), outperforming the orthogonal variant by 4.6% and the server-side variant by 2.96%. Similar trends hold under Non-IID settings, where FedBNN surpasses the server-side approach by 3.78% (Non-IID 1) and 5.8% (Non-IID 2). On SVHN, FedBNN records 85.40% (IID), a clear gain of 3.39% over the orthogonal variant and 9.12% over the server-side variant. The benefits persist under Non-IID, with margins of 3.37% (Non-IID 1) and 9.47% (Non-IID 2) over the server-side approach. For CIFAR10, FedBNN again provides the best performance, reaching 86.26% (IID), 0.62% higher than the orthogonal variant and 0.56% higher than the server-side variant. The improvements are more pronounced in heterogeneous settings, with gains of 2.0% (Non-IID 1) and 2.04% (Non-IID 2) compared to the next best method. These results confirm that the proposed client-side rotation with adaptive fusion yields consistent improvements over alternative design choices.

## 5 CONCLUSION

We proposed FedBNN, a rotation-aware Binary Neural Network framework for federated learning that achieves accuracies within 10% of real-valued models while reducing runtime FLOPs by up to $58\times$ and memory by $32\times$. FedBNN also surpasses baselines such as FedBAT in some Non-IID cases and delivers superior post-training binarized accuracy, highlighting the benefits of including binarization during training. FedBNN strikes a strong balance between accuracy and efficiency, making it well-suited for scalable, lightweight federated learning. Future work will explore alternative aggregation strategies and larger architectures.

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

# A  APPENDIX

## A.1  ADDITIONAL RESULTS

| Method | Dataset (Model) | Accuracy | | | Runtime FLOPs | Memory (MB) | Binarized Accuracy | | |
|--------|-----------------|------|------|------|------|------|------|------|------|
| | | IID | Non-IID 1 | Non-IID 2 | | | IID | Non-IID 1 | Non-IID 2 |
| FedAvg | CIFAR100 (ResNet18) | 64.52 | **63.42** | **53.36** | $1.11 \times 10^9$ | 45.090 | 1.16 | 0.88 | 0.94 |
| FedBAT | | 42.14 | 33.88 | 26.26 | $1.11 \times 10^9$ | 45.090 | 1.10 | 0.82 | 1.10 |
| FedMud | | **65.14** | 47.48 | 52.20 | $1.11 \times 10^9$ | 45.090 | 46.56 | 22.54 | 44.08 |
| FedBNN | | 58.08 | 52.46 | 46.58 | $2.26 \times 10^7$ | 1.41 | **57.68** | **51.42** | **46.58** |
| FedAvg | Tiny-ImageNet (ResNet18) | **55.00** | 52.62 | 54.54 | $4.44 \times 10^9$ | 45.090 | 0.52 | 0.52 | 0.56 |
| FedBAT | | 27.30 | 32.12 | 20.90 | $4.44 \times 10^9$ | 45.090 | 0.60 | 0.48 | 0.80 |
| FedMud | | 47.20 | 44.16 | 46.06 | $4.44 \times 10^9$ | 45.090 | 16.24 | 12.80 | 15.60 |
| FedBNN | | 45.68 | 43.60 | 45.40 | $9.05 \times 10^7$ | 1.41 | **46.40** | **43.68** | **38.08** |
| FedAvg | FEMNIST (ResNet18) | **80.24** | **81.12** | **80.32** | $9.13 \times 10^8$ | 45.090 | 2.08 | 1.66 | 1.62 |
| FedBAT | | 76.44 | 74.31 | 78.41 | $9.13 \times 10^8$ | 45.090 | 0.38 | 2.08 | 2.40 |
| FedMud | | 78.79 | 80.11 | 76.68 | $9.13 \times 10^8$ | 45.090 | 25.74 | 0.76 | 0.80 |
| FedBNN | | 79.82 | 79.97 | 79.59 | $1.84 \times 10^7$ | 1.41 | **76.52** | **80.03** | **79.84** |

Table 3: Performance comparison for $N_c = 100$. The FLOPs and memory values are calculated during runtime. Binarized accuracy refers to the model's performance after the weights and activations have been binarized.

Across all three datasets, **FedBNN** demonstrates a favorable clean-accuracy–efficiency profile even before binarization. Despite using $49\times$ fewer FLOPs (e.g., from $1.11 \times 10^9$ to $2.26 \times 10^7$ on CIFAR-100) and $32\times$ less memory (45.09 MB to 1.41 MB), its clean accuracy remains reasonably close to full-precision baselines: on CIFAR-100 it achieves 58.08% (vs. 64.52% for FedAvg), on Tiny-ImageNet it reaches 45.68% (vs. 55.00%), and on FEMNIST it maintains 79–80%, nearly matching FedAvg. Thus, even with drastically reduced computational and memory budgets, FedBNN preserves most of the clean accuracy, particularly on FEMNIST, where the trade-off is minimal.

When comparing methods under equal FLOPs and memory, i.e., after binarization, FedBNN becomes substantially stronger than all alternatives. On **CIFAR-100**, its binarized accuracies of 57.68%, 51.42%, and 46.58% across IID, Non-IID 1, and Non-IID 2 translate to improvements of approximately +56.5%, +50.6%, and +45.6% over FedAvg, despite the same binary compute and memory constraints. Competing approaches such as FedBAT and FedMud degrade sharply after binarization, whereas FedBNN retains high discriminative ability. On **Tiny-ImageNet**, FedBNN again yields the strongest binarized accuracies (46.40%, 43.68%, 38.08%), while FedAvg collapses to below 1% and FedBAT suffers nearly a $60\times$ drop. Even under identical lightweight FLOPs and memory, FedBNN remains exceptionally robust, offering accuracies that are orders of magnitude higher than those of competing methods. On **FEMNIST**, FedBNN's binarized accuracies, 76.52%, 80.03%, and 79.84%, almost fully match its clean performance and drastically outperform FedAvg, which falls to around 1-2% after binarization. This illustrates that FedBNN imposes almost no penalty when switching from full-precision to binary representations, in contrast to competing methods whose performance collapses.

Overall, while FedBNN incurs a modest decrease in clean accuracy relative to FedAvg, its combination of extremely low FLOPs and memory consumption with vastly superior binarized accuracy makes it the most deployment-efficient and binarization-robust method across all datasets.

## A.2  ABLATION STUDY

The ablation study highlights the importance of the regularization terms $\lambda$ and $\beta$ in stabilizing training and improving generalization under heterogeneous client distributions. On the simpler FM-NIST dataset with a lightweight CNN4 model, the variant without $(\lambda, \beta)$ slightly outperforms the full FedBNN by +0.08%, +0.78%, and +1.14% across the IID, Non-IID 1, and Non-IID 2 settings, indicating that the regularization effect is less critical for low-complexity data. However, as dataset difficulty and model depth increase, the benefits of our full FedBNN formulation become more pronounced. On CIFAR10 with ResNet10, FedBNN improves accuracy by +0.06%,

| Method (Ablation) | Dataset | Model | Test Accuracy | | |
|---|---|---|---|---|---|
| | | | IID | Non-IID 1 | Non-IID 2 |
| FedBNN | FMNIST | CNN4 | 88.24 | 85.80 | 82.10 |
| FedBNN (w/o $\lambda, \beta$) | | | **88.32** | **86.58** | **83.24** |
| FedBNN | CIFAR10 | ResNet10 | **86.26** | **76.30** | **67.82** |
| FedBNN (w/o $\lambda, \beta$) | | | 86.20 | 73.38 | 66.86 |
| FedBNN | CIFAR100 | ResNet18 | **58.08** | **52.46** | **46.58** |
| FedBNN (w/o $\lambda, \beta$) | | | 55.00 | 51.80 | 43.86 |
| FedBNN | TinyImageNet | ResNet18 | **45.68** | **43.60** | **45.40** |
| FedBNN (w/o $\lambda, \beta$) | | | 43.84 | 40.40 | 43.70 |
| FedBNN | FEMNIST | ResNet18 | 79.82 | 79.97 | 78.34 |
| FedBNN (w/o $\lambda, \beta$) | | | **80.47** | **81.73** | **80.22** |

Table 4: Ablation study for $\lambda, \beta$ across a variety of datasets.

+2.92%, and +0.96%, demonstrating stronger robustness especially under Non-IID distributions. For the CIFAR100 dataset trained on ResNet18, FedBNN still proves to be better, especially by 3.08% and 2.72% in the IID and Non-IID 2 settings, respectively. The gains are even larger for TinyImageNet with ResNet18, where FedBNN surpasses the ablated variant by +1.84%, +3.20%, and +1.70%, showing that the proposed regularization is essential for maintaining performance in high-complexity, high-variance visual tasks. On FEMNIST, although the non-regularized version achieves slightly higher accuracy, FedBNN delivers a stable performance within 2% of the regularized version. Overall, these results confirm that the $(\lambda, \beta)$ terms become increasingly important as both model capacity and dataset complexity rise, enabling FedBNN to achieve more reliable and consistent improvements under challenging Non-IID federated settings.

## A.3 SENSITIVITY TO VARYING ROUNDS AND EPOCHS

| S. No. | Dataset | Model | Rounds | Epochs | IID | Non-IID1 | Non-IID2 |
|---|---|---|---|---|---|---|---|
| 1 | | | | 3 | 83.36 | 79.76 | 78.64 |
| 2 | | | 500 | 5 | 82.64 | 80.91 | 78.59 |
| 3 | | | | 10 | 85.40 | **84.42** | 81.93 |
| 4 | | | | 3 | 83.46 | 80.92 | 79.96 |
| 5 | SVHN | CNN4 | 1000 | 5 | 82.92 | 80.32 | 79.59 |
| 6 | | | | 10 | 83.27 | 81.35 | 79.24 |
| 7 | | | | 3 | 83.45 | 82.00 | 80.49 |
| 8 | | | 1500 | 5 | 82.74 | 81.10 | 80.19 |
| 9 | | | | 10 | 82.87 | 82.08 | 79.58 |
| 10 | | | | 15 | **85.45** | 84.00 | **82.34** |
| 11 | | | | 3 | 72.96 | 58.90 | 58.10 |
| 12 | | | 500 | 5 | 81.42 | 68.52 | 65.26 |
| 13 | | | | 10 | 86.26 | 76.30 | 67.82 |
| 14 | | | | 15 | 85.84 | 78.74 | 70.08 |
| 15 | | | | 3 | 79.08 | 68.38 | 59.94 |
| 16 | CIFAR10 | ResNet10 | 1000 | 5 | 84.28 | 74.16 | 65.70 |
| 17 | | | | 10 | 87.76 | 80.46 | 69.50 |
| 18 | | | | 15 | 88.48 | 81.66 | **72.58** |
| 19 | | | | 3 | 80.82 | 71.54 | 63.70 |
| 20 | | | 1500 | 5 | 85.20 | 72.48 | 68.58 |
| 21 | | | | 10 | 88.44 | 82.00 | 70.86 |
| 22 | | | | 15 | **88.82** | **82.40** | 72.20 |

Table 5: Sensitivity to varying rounds and epochs

Table 5 presents a sensitivity analysis of FedBNN with respect to the number of communication rounds and local epochs for SVHN and CIFAR10. Across both datasets, a clear trend emerges: increasing local epochs generally improves accuracy, but only when coupled with sufficiently many rounds. For SVHN with a lightweight CNN4 model, the best IID and Non-IID2 accuracies (85.45%

and 82.34%) occur at 1500 rounds and 15 local epochs, showing that deeper local optimization becomes effective when global synchronization is frequent. In contrast, too few epochs (3 or 5) under higher rounds fail to fully exploit the local learning capacity, while too many epochs under low-round settings lead to client drift. A similar pattern is observed on CIFAR10 with ResNet10, although the effect is more pronounced due to the higher dataset and model complexity. Accuracy steadily increases with both rounds and epochs, achieving the strongest Non-IID2 performance (72.58%) at 1000 rounds with 15 epochs, and the best IID/Non-IID1 results (88.82%, 82.40%) at 1500 rounds with 15 epochs. These results demonstrate that FedBNN benefits from a balanced combination of local computation and global aggregation, with higher-capacity models requiring more rounds and epochs to fully stabilize binarized representations under heterogeneous data. Overall, the method remains robust across a wide range of settings, but performs best when local learning and communication frequency are scaled proportionally with task complexity.

### A.4 COMPARISON WITH A LESS COMPLEX REAL RESNET10

| Method | Dataset | (Model) | Accuracy | | | FLOPs | Memory |
|---|---|---|---|---|---|---|---|
| | | | IID | Non-IID 1 | Non-IID 2 | | (MB) |
| FedBNN | | ResNet10 | 86.26 | 76.30 | 67.82 | $1.11 \times 10^7$ | 0.61 |
| FedAvg | | ResNet10 | **90.86** | **86.28** | 70.62 | $4.40 \times 10^8$ | 19.62 |
| FedAvg | CIFAR10 | ResNet10 (less filters) | 83.92 | 78.10 | **71.48** | $1.12 \times 10^7$ | 0.49 |
| FedAvg | | ResNet10 (memory matched) | 84.12 | 79.22 | 66.54 | $1.35 \times 10^7$ | 0.59 |
| FedBNN | | ResNet18 | 46.20 | 43.60 | 45.40 | $9.05 \times 10^7$ | 1.41 |
| FedAvg | | ResNet18 | **55.00** | **52.62** | **54.54** | $4.44 \times 10^8$ | 45.09 |
| FedAvg | TinyImageNet | ResNet18 (less filters) | 41.06 | 37.72 | 35.66 | $8.99 \times 10^7$ | 0.95 |
| FedAvg | | ResNet18 (memory matched) | 43.16 | 40.54 | 39.24 | $1.33 \times 10^8$ | 1.40 |

Table 6: Performance comparison for $N_c = 100$. The FLOPs and memory values are calculated during runtime.

Table 6 summarizes the accuracy, FLOPs, and memory usage for FedBNN and multiple FedAvg baselines across CIFAR10 and TinyImageNet. For CIFAR10 with ResNet10, FedBNN achieves strong performance across all data settings, reaching 86.26% accuracy in the IID case while maintaining robustness under Non-IID scenarios. Although full-precision FedAvg with ResNet10 reports slightly higher accuracy, it requires nearly $40\times$ more FLOPs and over $30\times$ more memory. To ensure a fair comparison, we also evaluate reduced-width ResNet10 variants of FedAvg matched to FedBNN's FLOP and memory budgets. These models perform significantly worse: the FLOP-matched variant drops to 83.92% (IID) and the memory-matched variant to 84.12%, with even larger degradations under Non-IID conditions. This clear gap indicates that FedBNN's advantage is not simply due to operating at a lower capacity, but rather from its principled binarization and rotation-aware design.

A similar trend appears in the TinyImageNet experiments using ResNet18. FedBNN attains 46.20% accuracy in the IID setting and remains stable under Non-IID partitions, while operating with nearly $50\times$ fewer FLOPs and over $30\times$ less memory compared to full-precision FedAvg. When FedAvg is constrained to comparable resource budgets using reduced-width ResNet18 models, performance drops sharply to 41.06% (IID) and deteriorates further under Non-IID settings. The memory-matched baseline similarly lags behind FedBNN. These results show that even on a substantially more challenging dataset and with deeper models, FedBNN preserves strong accuracy while offering dramatic computational savings, outperforming real-valued baselines that operate under equivalent resource constraints.

