# OpenReview forum: "Federated Learning with Binary Neural Networks: Competitive Accuracy at a Fraction of the Cost"
_ICLR.cc/2026/Conference — Submitted to ICLR 2026_

### Official Review · Reviewer_yw3J · 2025-10-28

**Soundness:** 2
**Presentation:** 2
**Contribution:** 3
**Rating:** 4
**Confidence:** 3

**Summary:**

This paper proposes FedBNN, a Rotation-Aware Binary Neural Network framework for Federated Learning (FL). It aims to address the significant computational and storage burdens associated with training and inference on edge devices within the FL paradigm. FedBNN directly learns binary weights on the client side. To mitigate angular bias induced by binarization, it introduces trainable rotation matrices and an adaptive mixing weight strategy. Furthermore, it enhances gradient propagation during backpropagation through a training-aware approximation function. Experiments conducted on datasets including FMNIST, SVHN, and CIFAR-10 demonstrate that FedBNN achieves within 10% of its real-valued counterparts while reducing runtime FLOPs by approximately 40–58 times and memory footprint by 32 times. Ablation studies are also presented to analyze the impact of rotation matrix aggregation and initialization strategies.

**Strengths:**

1.Federated Extension of Rotational Binary Neural Networks: Adapting rotational binary neural networks to the federated learning framework, this work extends the centralized rotational optimization proposed by Lin et al. (2020) to federated learning scenarios. By introducing trainable rotation matrices and a federation-aware weighting mechanism, the angular deviation caused by binarization is effectively mitigated, balancing local optimization with global consistency.

2.Significant Computational Cost Reduction: Achieves 40–58 times lower runtime FLOPs and 32 times reduced memory usage, validating edge deployment potential.

3.Comprehensive Experiments: Evaluated on FMNIST, SVHN, CIFAR-10 under IID/non-IID scenarios, with comparisons to FedAvg/FedBAT/FedMud. in computational load, storage cost, and accuracy.

**Weaknesses:**

1.Limited theoretical analysis: Convergence proof of FedBNN in federated scenarios is not provided (compared to the theoretical analysis of FedAvg). The impact of rotational optimization on convergence speed also needs to be quantified (e.g., upper bound of convergence rounds).

2.Insufficient dataset complexity: Validation is only conducted on small-to-medium-scale datasets (FMNIST/SVHN/CIFAR-10), lacking verification on more complex or high-dimensional datasets (e.g., CIFAR-100 or ImageNet).

3.Inadequate model complexity: FedBNN is experimentally validated only on ResNet-10 and a small CNN, with unknown optimization effects on deeper models.

4.Lack of quantitative analysis on communication cost: Although the paper claims significant savings in inference and storage, transmitting rotation matrices may introduce additional communication overhead. The experiment also fails to compare the communication cost of FedBNN with FedAvg/FedBAT/FedMud.

**Questions:**

1.Convergence analysis: It is recommended to supplement the theoretical convergence analysis of FedBNN in federated scenarios. For example, by comparing with the existing theoretical framework of FedAvg, further quantifying the impact of rotational optimization on convergence speed (e.g., convergence round boundaries) to enhance the theoretical support of the method.

2.Communication overhead: Transmitting rotation matrices may introduce additional communication costs. It is suggested to supplement the quantitative analysis of this overhead and compare it with the communication costs of methods like FedAvg and FedBAT to comprehensively evaluate overall performance.

3.Scalability validation: Current experiments are mainly based on small-to-medium-scale datasets and shallow models. It is recommended to further validate on more complex datasets (e.g., CIFAR-100, ImageNet) and deeper networks (e.g., ResNet-18, MobileNetV3) to demonstrate the method’s generality.

4.Practical deployment testing: It is suggested to supplement deployment tests in real edge device environments to verify the feasibility and efficiency of the method under actual hardware conditions.

---

> ### Author Response · Authors · 2025-11-27
> **Tables for Extra Simulations**
>
> # Reviewer 4: Tables
>
> ## Table 1: Performance Comparison {#tab-additional4}
>
> **Performance comparison for N_c = 100. The FLOPs and memory values are calculated during runtime. Binarized accuracy refers to the model's performance after the weights and activations have been binarized.**
>
> | Method | Dataset (Model) | Accuracy (IID) | Accuracy (Non-IID 1) | Accuracy (Non-IID 2) | Runtime FLOPs | Memory (MB) | Binarized Accuracy (IID) | Binarized Accuracy (Non-IID 1) | Binarized Accuracy (Non-IID 2) |
> |:-------|:---------------|:---------------|:---------------------|:---------------------|:--------------|:------------|:------------------------|:-------------------------------|:-------------------------------|
> | FedAvg | CIFAR100 | 64.52 | **63.42** | **53.36** | 1.11 × 10⁹ | 45.090 | 1.16 | 0.88 | 0.94 |
> | FedBAT | CIFAR100 (ResNet18) | 42.14 | 33.88 | 26.26 | 1.11 × 10⁹ | 45.090 | 1.10 | 0.82 | 1.10 |
> | FedMud | CIFAR100 (ResNet18) | **65.14** | 47.48 | 52.20 | 1.11 × 10⁹ | 45.090 | 46.56 | 22.54 | 44.08 |
> | FedBNN | CIFAR100 (ResNet18) | 58.08 | 52.46 | 46.58 | 2.26 × 10⁷ | 1.41 | **57.68** | **51.42** | **46.58** |
> | FedAvg | TinyImageNet (ResNet18) | **55.00** | **52.62** | **54.54** | 4.44 × 10⁹ | 45.090 | 0.52 | 0.52 | 0.56 |
> | FedBAT | TinyImageNet (ResNet18) | 27.30 | 32.12 | 20.90 | 4.44 × 10⁹ | 45.090 | 0.60 | 0.48 | 0.80 |
> | FedMud | TinyImageNet (ResNet18) | 47.20 | 44.16 | 46.06 | 4.44 × 10⁹ | 45.090 | 16.24 | 12.80 | 15.60 |
> | FedBNN | TinyImageNet (ResNet18) | 45.68 | 43.60 | 45.40 | 9.05 × 10⁷ | 1.41 | **46.40** | **43.68** | **38.08** |
> | FedAvg | FEMNIST (ResNet18) | **80.24** | **81.12** | **80.32** | 9.13 × 10⁸ | 45.090 | 2.08 | 1.66 | 1.62 |
> | FedBAT | FEMNIST (ResNet18) | 76.44 | 74.31 | 78.41 | 9.13 × 10⁸ | 45.090 | 0.38 | 2.08 | 2.40 |
> | FedMud | FEMNIST (ResNet18) | 78.79 | 80.11 | 76.68 | 9.13 × 10⁸ | 45.090 | 25.74 | 0.76 | 0.80 |
> | FedBNN | FEMNIST (ResNet18) | 79.82 | 79.97 | 79.59 | 1.84 × 10⁷ | 1.41 | **76.52** | **80.03** | **79.84** |
>
> ## Table 2: Sensitivity to Rounds and Epochs {#rounds-epochs4}
>
> | S. No. | Dataset | Model | Rounds | Epochs | IID | Non-IID1 | Non-IID2 |
> |:-------|:--------|:------|:-------|:-------|:---|:---------|:---------|
> | 1 | | | 500 | 3 | 83.36 | 79.76 | 78.64 |
> | 2 | | | 500 | 5 | 82.64 | 80.91 | 78.59 |
> | 3 | | | 500 | 10 | 85.40 | **84.42** | 81.93 |
> | 4 | | | 1000 | 3 | 83.46 | 80.92 | 79.96 |
> | 5 | SVHN | CNN4 | 1000 | 5 | 82.92 | 80.32 | 79.59 |
> | 6 | | | 1000 | 10 | 83.27 | 81.35 | 79.24 |
> | 7 | | | 1500 | 3 | 83.45 | 82.00 | 80.49 |
> | 8 | | | 1500 | 5 | 82.74 | 81.10 | 80.19 |
> | 9 | | | 1500 | 10 | 82.87 | 82.08 | 79.58 |
> | 10 | | | 1500 | 15 | **85.45** | 84.00 | **82.34** |
> | 11 | | | 500 | 3 | 72.96 | 58.90 | 58.10 |
> | 12 | | | 500 | 5 | 81.42 | 68.52 | 65.26 |
> | 13 | | | 500 | 10 | 86.26 | 76.30 | 67.82 |
> | 14 | | | 500 | 15 | 85.84 | 78.74 | 70.08 |
> | 15 | | | 1000 | 3 | 79.08 | 68.38 | 59.94 |
> | 16 | CIFAR10 | ResNet10 | 1000 | 5 | 84.28 | 74.16 | 65.70 |
> | 17 | | | 1000 | 10 | 87.76 | 80.46 | 69.50 |
> | 18 | | | 1000 | 15 | 88.48 | 81.66 | **72.58** |
> | 19 | | | 1500 | 3 | 80.82 | 71.54 | 63.70 |
> | 20 | | | 1500 | 5 | 85.20 | 72.48 | 68.58 |
> | 21 | | | 1500 | 10 | 88.44 | 82.00 | 70.86 |
> | 22 | | | 1500 | 15 | **88.82** | **82.40** | 72.20 |

---

> > ### Author Response · Authors · 2025-11-27
> > **Response to Weakness 2, 3, 4**
> >
> > - **Response to Weakness 2, 3**: We acknowledge the reviewer's interest in the applicability of our method to more complex architectures and datasets. Based on the reviewer's comments, we have benchmarked our proposed FedBNN method using the ResNet18 model for the CIFAR100, TinyImageNet, and FEMNIST datasets. Please refer to [Table 1](reviewer4_tables.md#tab-additional4). On CIFAR-100, we observe that FedBNN outperforms FedBAT by at least 15% across different strategies, while achieving 32× savings in runtime memory and a 49× reduction in runtime FLOPs. On TinyImageNet, FedBNN outperforms FedBAT by 18.9% in IID settings and 24.5% in non-IID settings, while achieving similar savings in complexity. Additionally, on the FEMNIST dataset, FedBNN outperforms FedBAT while matching the performance of FedAVG, and has significantly lower runtime complexity. These results further validate the performance and competitiveness of our approach on current benchmarks.
> >
> > - **Response to Weakness 4**: We appreciate the reviewer's concern. FedBAT indeed provides substantial communication benefits by transmitting binary model updates, resulting in an approximate 32× reduction in payload compared to algorithms that send full-precision weights, such as FedAvg or FedMud. This reduction directly lowers the uplink cost per FL round. However, with modern wireless technologies such as 5G, mmWave, and emerging 6G systems, communication bandwidth has increased to the point where the cost of transmitting even full-precision updates is becoming increasingly insignificant. In practical FL deployments, especially in cross-silo federated learning scenarios such as collaborations across hospitals, institutions, or enterprise clusters, as well as many cross-device settings, the training is typically performed on clients equipped with sufficient compute resources such as GPUs or edge accelerators (e.g., hospital servers, campus clusters, enterprise gateways). Thus, the temporary increase in computation during training is manageable.
> >
> >   Beyond communication, it is important to note that FedBNN provides a different and often more impactful advantage: substantially reduced runtime computational complexity. Unlike FedBAT, which trains a full-precision model locally and only binarizes the updates before transmission, FedBNN trains a fully binary model throughout both the forward and backward passes. This results in dramatically lower FLOPs and memory requirements during local training, leading to faster on-device computation and lower energy consumption. As shown in our experiments, FedBNN reduces runtime FLOPs by approximately 96–98% and memory usage by nearly 97% compared to full-precision methods, including FedBAT. Thus, while FedBAT's communication savings may be less impactful in the era of high-bandwidth wireless systems, FedBNN directly benefits client-side efficiency by reducing computational cost, an aspect that remains critical due to the limited battery, processing capability, and thermal constraints of mobile and edge devices.
> >
> >   In summary, although FedBAT excels in communication compression, FedBNN provides significantly greater runtime efficiency by operating entirely in the binary domain, making it more suitable for real-world, resource-constrained federated learning settings.

---

> > > ### Author Response · Authors · 2025-11-27
> > > **Response to Weakness 1 and Questions 1, 2, 3, 4**
> > >
> > > - **Response to Weakness 1 and Question 1**: We thank the reviewer for asking for a convergence proof. Formal convergence analysis for FL is typically demonstrated for simpler ML models, such as the Perceptron, Linear Regression, and SVM. However, demonstrating convergence for deep neural networks is inherently challenging due to the presence of nonlinearity.
> > >
> > >   To further investigate convergence, we performed additional experiments by varying the number of rounds and local epochs for the SVHN and CIFAR-10 datasets, as shown in [Table 2](reviewer4_tables.md#rounds-epochs4). These experiments demonstrate a clear and consistent trend: increasing the number of rounds and local epochs leads to stable performance improvements, indicating continued optimization progress rather than stagnation especially for the more complex CIFAR10 dataset. Upon increasing the number of rounds and local epochs, we observe a performance increase for both datasets, particularly 0.41% in Non-IID 2 for SVHN and 6.1% in Non-IID 1 for CIFAR10. While not a formal proof, this empirical analysis provides strong evidence that the model continues to benefit from additional optimization steps, supporting the claim that FedBNN exhibits stable and improving convergence behavior as training progresses, especially on more complex datasets.
> > >
> > >   Additionally, beyond the simple models evaluated in earlier work, we further investigate the convergence behaviour of the proposed method using more complex architectures on challenging datasets, such as Tiny ImageNet, CIFAR-100, and FEMNIST. These additional experiments provide stronger evidence of stable and consistent convergence under more realistic and demanding conditions. As shown in [Table 1](reviewer4_tables.md#tab-additional4), on CIFAR-100, we observe that FedBNN outperforms FedBAT by at least 15% across different strategies, while achieving 32× savings in runtime memory and a 49× reduction in runtime FLOPs. On TinyImageNet, FedBNN outperforms FedBAT by 18.9% in IID and 24.5% in non-IID settings. Additionally, on the FEMNIST dataset, FedBNN outperforms FedBAT while matching the performance of FedAVG, and has significantly lower runtime complexity.
> > >
> > >   Also, as mentioned in Section 4.1, all results in TABLE 1 and TABLE 2 of the original paper are the submitted manuscript for 500 rounds and 10 epochs (5 epochs for FMNIST) across all methods being compared.
> > >
> > > - **Response to Question 2**: We computed the additional communication required to transmit the rotation matrices R₁ and R₂ for both architectures (ResNet10 and CNN4). The results are as follows:
> > >
> > >   1. **ResNet10:**
> > >      Uplink real-valued weights = 156,719,104 bits
> > >      Rotation matrices $R_1, R_2$ = 873,472 bits
> > >      Overhead = **0.5%**
> > >
> > >   2. **CNN4:**
> > >      Uplink real-valued weights = 12,477,440 bits
> > >      Rotation matrices $R_1, R_2$ = 130,240 bits
> > >      Overhead = **1%**
> > >
> > >   Since the increase in the number of bits transmitted to the server is ≤ 1% for both architectures, the communication overhead introduced by R₁ and R₂ is insignificant.
> > >
> > > - **Response to Question 3**: We acknowledge the reviewer's interest in the applicability of our method to more complex architectures and datasets. Based on the reviewer's comments, we have benchmarked our proposed FedBNN method using the ResNet18 model for the CIFAR100, TinyImageNet, and FEMNIST datasets. Please refer to [Table 1](reviewer4_tables.md#tab-additional4). On CIFAR-100, we observe that FedBNN outperforms FedBAT by at least 15% across different strategies, while achieving 32× savings in runtime memory and a 49× reduction in runtime FLOPs. On TinyImageNet, FedBNN outperforms FedBAT by 18.9% in IID settings and 24.5% in non-IID settings, while achieving similar savings in complexity. Additionally, on the FEMNIST dataset, FedBNN outperforms FedBAT while matching the performance of FedAVG, and has significantly lower runtime complexity. These results further validate the performance and competitiveness of our approach on current benchmarks.
> > >
> > > - **Response to Question 4**: We appreciate the suggestion regarding practical deployment tests on real edge devices. However, we are not from that specific application area, and conducting hardware-level deployment experiments is outside the scope of our current setup.
> > >
> > >   Nonetheless, prior works already provide strong evidence that Binary Neural Networks (BNNs) offer substantial computation and latency advantages when deployed on hardware compared to real-valued models. For example, BNNs replace costly floating-point MACs with bitwise XNOR and popcount operations, achieving 58× faster convolution operations, up to 32× memory reduction [1]. These results suggest that BNN-style models are well-suited for efficient on-device deployment.
> > >
> > > Reference:
> > > 1. Rastegari, Mohammad, et al. "Xnor-net: Imagenet classification using binary convolutional neural networks." European conference on computer vision. Cham: Springer International Publishing, 2016.

---

### Official Review · Reviewer_m1GA · 2025-10-30

**Soundness:** 2
**Presentation:** 2
**Contribution:** 2
**Rating:** 6
**Confidence:** 3

**Summary:**

This paper introduces **FedBNN**, a rotation-aware **Binary Neural Network (BNN)** framework for **Federated Learning (FL)** that achieves competitive accuracy with dramatically reduced computation and memory costs. Unlike conventional FL methods that rely on full-precision models. To mitigate quantization error, FedBNN applies trainable rotation matrices before binarization and introduces **federated-aware fusion** and **adaptive rotation adjustment** mechanisms that align local and global model directions under heterogeneous data. A training-aware approximation function further stabilizes gradient flow during optimization. Experiments show that FedBNN maintains accuracy within 5–10% of real-valued models while achieving superior efficiency, outperforming existing binarization-based FL methods such as FedBAT.

**Strengths:**

The paper demonstrates strong **originality** by being the first to integrate rotation-aware binary neural networks—originally designed for centralized settings—into federated learning, introducing a novel client-side fusion of local and global weights before rotation and adaptive correction terms (α, β) that jointly minimize quantization-induced angular bias and align with the server model. Its **technical quality** is high, featuring a well-motivated method (FedBNN), rigorous experiments across three datasets under IID and two levels of Non-IID heterogeneity, and comprehensive comparisons against state-of-the-art baselines like FedBAT and FedMud.

**Weaknesses:**

A key weakness of the paper is the lack of a fair comparison under **equal computational budgets**. The authors claim FedBNN achieves "competitive accuracy" by comparing its binary model directly against full-precision baselines like FedAvg, but on the performance, there is a significant gap. As a result, a smaller model with the same FLOPs may still achieve the same performance as FedBNN. Without evaluating a smaller real-valued model matched to FedBNN’s computational cost—such as a pruned or shallower CNN—the claimed accuracy trade-off remains misleading; the performance gap may stem more from model capacity disparity than binarization itself. Furthermore, the efficiency gains (FLOPs and memory) are **purely theoretical** and not validated on real edge hardware. Since XNOR+bitcount acceleration depends heavily on hardware support (e.g., ARM NEON or specialized accelerators), the absence of end-to-end latency, energy consumption, or inference speed measurements on actual devices leaves the practical deployability of FedBNN unsubstantiated. Including both equal-budget comparisons and real-device benchmarks would significantly strengthen the paper’s claims.

**Questions:**

1. **Fairness of Accuracy Comparison**:
   The paper compares FedBNN’s accuracy against full-precision FedAvg while highlighting massive FLOPs/memory savings. However, this comparison conflates *model capacity* with *quantization effects*. Could the authors provide results for a **smaller real-valued model** (e.g., a pruned or width-scaled CNN/ResNet) that matches FedBNN’s FLOPs or memory footprint? This would clarify whether the accuracy gap is due to binarization itself or simply reduced representational capacity.

2. **Real-Device Efficiency Validation**:
   All efficiency claims (FLOPs, memory) are theoretical. Could the authors provide **real-world latency or energy measurements** on an edge device (e.g., Raspberry Pi, smartphone) comparing FedBNN against a real-valued baseline? Without this, the practical deployability of the claimed 40–58× speedup remains speculative.

Addressing these points would significantly bolster the paper’s technical rigor and practical relevance.

---

> ### Author Response · Authors · 2025-12-02
> **Table 1: Performance Comparison of FedBNN with a FLOP/memory matched FedAvg ResNet10.**
>
> **Table 1: Performance Comparison of FedBNN with a FLOP/memory matched FedAvg ResNet10. N_c = 100. The FLOPs and memory values are calculated during runtime.**
>
> | Method     | Dataset          | (Model)                 | IID       | Non-IID 1 | Non-IID 2 | FLOPs            | Memory (MB) |
> | ---------- | ---------------- | ----------------------- | --------- | --------- | --------- | ---------------- | ----------- |
> | **FedBNN** |                  | ResNet10                | 86.26     | 76.30     | 67.82     | $1.11\times10^7$ | 0.61        |
> | **FedAvg** |                  | ResNet10                | **90.86** | **86.28** | 70.62     | $4.40\times10^8$ | 19.62       |
> | **FedAvg** | **CIFAR10**      | ResNet10 (less filters) | 83.92     | 78.10     | **71.48** | $1.12\times10^7$ | 0.49        |
> | **FedAvg** |                  | ResNet10 (mem matched)  | 84.12     | 79.22     | 66.54     | $1.35\times10^7$ | 0.59        |
> |||||||||||
> | **FedBNN** |                  | ResNet18                | 46.20     | 43.60     | 45.40     | $9.05\times10^7$ | 1.41        |
> | **FedAvg** |                  | ResNet18                | **55.00** | **52.62** | **54.54** | $4.44\times10^8$ | 45.09       |
> | **FedAvg** | **TinyImageNet** | ResNet18 (less filters) | 41.06     | 37.72     | 35.66     | $8.99\times10^7$ | 0.95        |
> | **FedAvg** |                  | ResNet18 (mem matched)  | 43.16     | 40.54     | 39.24     | $1.33\times10^8$ | 1.40        |
>
> - **Response to Question 1:** Table 1 summarizes the accuracy, FLOPs, and memory usage for FedBNN and multiple FedAvg baselines across CIFAR10 and TinyImageNet. For CIFAR10 with ResNet10, FedBNN achieves strong performance across all data settings, reaching $86.26\%$ accuracy in the IID case while maintaining robustness under Non-IID scenarios. Although full-precision FedAvg with ResNet10 reports slightly higher accuracy, it requires nearly $40\times$ more FLOPs and over $30\times$ more memory. To ensure a fair comparison, we also evaluate reduced-width ResNet10 variants of FedAvg matched to FedBNN’s FLOP and memory budgets. These models perform significantly worse: the FLOP-matched variant drops to $83.92\%$ (IID) and the memory-matched variant to $84.12\%$, with even larger degradations under Non-IID conditions. This clear gap indicates that FedBNN’s advantage is not simply due to operating at a lower capacity, but rather from its principled binarization and rotation-aware design.
>
>   A similar trend appears in the TinyImageNet experiments using ResNet18. FedBNN attains $46.20\%$ accuracy in the IID setting and remains stable under Non-IID partitions, while operating with nearly $50\times$ fewer FLOPs and over $30\times$ less memory compared to full-precision FedAvg. When FedAvg is constrained to comparable resource budgets using reduced-width ResNet18 models, performance drops sharply to $41.06\%$ (IID) and deteriorates further under Non-IID settings. The memory-matched baseline similarly lags behind FedBNN. These results show that even on a substantially more challenging dataset and with deeper models, FedBNN preserves strong accuracy while offering dramatic computational savings, outperforming real-valued baselines that operate under equivalent resource constraints.
>
> - **Response to Question 2**: We appreciate the suggestion regarding practical deployment tests on real edge devices. However, we are not from that specific application area, and conducting hardware-level deployment experiments is outside the scope of our current setup.
>
>   Nonetheless, prior works already provide strong evidence that Binary Neural Networks offer substantial computation and latency advantages when deployed on hardware compared to real-valued models. For example, BNNs replace costly floating-point MACs with bitwise XNOR and popcount operations, achieving 58× faster convolution operations, up to 32× memory reduction [1]. These results suggest that BNN-style models are well-suited for efficient on-device deployment.
>
> Reference:
> 1. Rastegari, Mohammad, et al. "Xnor-net: Imagenet classification using binary convolutional neural networks." European conference on computer vision. Cham: Springer International Publishing, 2016.

---

### Official Review · Reviewer_aihN · 2025-10-30

**Soundness:** 2
**Presentation:** 2
**Contribution:** 2
**Rating:** 2
**Confidence:** 4

**Summary:**

The paper studies federated learning on heterogeneous, resource-limited devices, focusing on reducing communication, computation, and memory overhead. The core idea is to train binarized versions of the model on devices while maintaining competitive accuracy. However, the contribution appears limited, as the proposed approach targets only CNN architectures, the performance analysis is restricted to relatively small models, and there is not proof of convergence.

**Strengths:**

The problem is important and has been studied extensively in recent years; efforts to reduce the resource footprint of federated learning are valuable.

**Weaknesses:**

1. The method is presented only for simple CNN models, with no discussion of whether it can extend to more complex architectures (e.g., LLMs).

2. The evaluation is restricted to relatively small models: four-layer CNNs for FEMNIST and SVHN, and a ResNet-10 for CIFAR-10.

3. There is no convergence proof; thus, convergence is assessed only empirically—and those experiments are limited to very simple models.

**Questions:**

1. Can the proposed method be extended to more complex architectures, such as LLMs or ResNet-50?

2. Is there any guarantee of training convergence? The current evaluation is limited to very simple models, so such a conclusion cannot be drawn from these results.

3. Could the authors provide more details on the convergence behavior of FedBNN? Are the results in Tables 1 and 2 reported for the same number of rounds, or does FedBNN require more rounds to converge?

4. Why are the baselines limited to this specific set? What about other state-of-the-art methods, such as FedProx?

---

> ### Author Response · Authors · 2025-11-27
> **Tables for Extra Simulations**
>
> ## Table 1: Performance Comparison {#tab-additional2}
>
> **Performance comparison for N_c = 100. The FLOPs and memory values are calculated during runtime. Binarized accuracy refers to the model's performance after the weights and activations have been binarized.**
>
> | Method | Dataset (Model) | Accuracy (IID) | Accuracy (Non-IID 1) | Accuracy (Non-IID 2) | Runtime FLOPs | Memory (MB) | Binarized Accuracy (IID) | Binarized Accuracy (Non-IID 1) | Binarized Accuracy (Non-IID 2) |
> |:-------|:---------------|:---------------|:---------------------|:---------------------|:--------------|:------------|:------------------------|:-------------------------------|:-------------------------------|
> | FedAvg | CIFAR100 | 64.52 | **63.42** | **53.36** | 1.11 × 10⁹ | 45.090 | 1.16 | 0.88 | 0.94 |
> | FedBAT | CIFAR100 (ResNet18) | 42.14 | 33.88 | 26.26 | 1.11 × 10⁹ | 45.090 | 1.10 | 0.82 | 1.10 |
> | FedMud | CIFAR100 (ResNet18) | **65.14** | 47.48 | 52.20 | 1.11 × 10⁹ | 45.090 | 46.56 | 22.54 | 44.08 |
> | FedBNN | CIFAR100 (ResNet18) | 58.08 | 52.46 | 46.58 | 2.26 × 10⁷ | 1.41 | **57.68** | **51.42** | **46.58** |
> | FedAvg | TinyImageNet (ResNet18) | **55.00** | **52.62** | **54.54** | 4.44 × 10⁹ | 45.090 | 0.52 | 0.52 | 0.56 |
> | FedBAT | TinyImageNet (ResNet18) | 27.30 | 32.12 | 20.90 | 4.44 × 10⁹ | 45.090 | 0.60 | 0.48 | 0.80 |
> | FedMud | TinyImageNet (ResNet18) | 47.20 | 44.16 | 46.06 | 4.44 × 10⁹ | 45.090 | 16.24 | 12.80 | 15.60 |
> | FedBNN | TinyImageNet (ResNet18) | 45.68 | 43.60 | 45.40 | 9.05 × 10⁷ | 1.41 | **46.40** | **43.68** | **38.08** |
> | FedAvg | FEMNIST (ResNet18) | **80.24** | **81.12** | **80.32** | 9.13 × 10⁸ | 45.090 | 2.08 | 1.66 | 1.62 |
> | FedBAT | FEMNIST (ResNet18) | 76.44 | 74.31 | 78.41 | 9.13 × 10⁸ | 45.090 | 0.38 | 2.08 | 2.40 |
> | FedMud | FEMNIST (ResNet18) | 78.79 | 80.11 | 76.68 | 9.13 × 10⁸ | 45.090 | 25.74 | 0.76 | 0.80 |
> | FedBNN | FEMNIST (ResNet18) | 79.82 | 79.97 | 79.59 | 1.84 × 10⁷ | 1.41 | **76.52** | **80.03** | **79.84** |
> ## Table 2: Sensitivity to Rounds and Epochs {#rounds-epochs2}
>
> | S. No. | Dataset | Model | Rounds | Epochs | IID | Non-IID1 | Non-IID2 |
> |:-------|:--------|:------|:-------|:-------|:---|:---------|:---------|
> | 1 | | | 500 | 3 | 83.36 | 79.76 | 78.64 |
> | 2 | | | 500 | 5 | 82.64 | 80.91 | 78.59 |
> | 3 | | | 500 | 10 | 85.40 | **84.42** | 81.93 |
> | 4 | | | 1000 | 3 | 83.46 | 80.92 | 79.96 |
> | 5 | SVHN | CNN4 | 1000 | 5 | 82.92 | 80.32 | 79.59 |
> | 6 | | | 1000 | 10 | 83.27 | 81.35 | 79.24 |
> | 7 | | | 1500 | 3 | 83.45 | 82.00 | 80.49 |
> | 8 | | | 1500 | 5 | 82.74 | 81.10 | 80.19 |
> | 9 | | | 1500 | 10 | 82.87 | 82.08 | 79.58 |
> | 10 | | | 1500 | 15 | **85.45** | 84.00 | **82.34** |
> |||||||||
> | 11 | | | 500 | 3 | 72.96 | 58.90 | 58.10 |
> | 12 | | | 500 | 5 | 81.42 | 68.52 | 65.26 |
> | 13 | | | 500 | 10 | 86.26 | 76.30 | 67.82 |
> | 14 | | | 500 | 15 | 85.84 | 78.74 | 70.08 |
> | 15 | | | 1000 | 3 | 79.08 | 68.38 | 59.94 |
> | 16 | CIFAR10 | ResNet10 | 1000 | 5 | 84.28 | 74.16 | 65.70 |
> | 17 | | | 1000 | 10 | 87.76 | 80.46 | 69.50 |
> | 18 | | | 1000 | 15 | 88.48 | 81.66 | **72.58** |
> | 19 | | | 1500 | 3 | 80.82 | 71.54 | 63.70 |
> | 20 | | | 1500 | 5 | 85.20 | 72.48 | 68.58 |
> | 21 | | | 1500 | 10 | 88.44 | 82.00 | 70.86 |
> | 22 | | | 1500 | 15 | **88.82** | **82.40** | 72.20 |

---

> ### Author Response · Authors · 2025-11-27
> **Response to Weakness 1,2 and 3 and Question 1**
>
> - **Response Weakness 1,2**: We acknowledge the reviewer's interest in the applicability of our method to more complex architectures such as LLMs. However, it is important to recognize that training large-scale models in federated settings is fundamentally constrained by device- and system-level resources. In practical FL deployments, whether cross-device or cross-silo, the majority of clients cannot support the memory footprint, communication bandwidth, or computational requirements needed for training LLMs or similarly massive architectures. Especially, in our experiments, the computational requirements for training 10 such LLMs will not be feasible. This is precisely why most existing FL research evaluates on lightweight CNNs or ResNets, which reflect the realistic capabilities of edge devices, hospital servers, or enterprise gateways. Within these constraints, our proposed FedBNN method directly targets the core challenges of resource-efficient federated inference, and we have extended the experiments to ResNet18 model for CIFAR100, TinyImageNet and FEMNIST datasets. Please refer to [Table 1](#tab-additional2). On CIFAR-100, we observe that FedBNN outperforms FedBAT by at least 15% across different strategies, while achieving 32× savings in runtime memory and a 49× reduction in runtime FLOPs. On TinyImageNet, FedBNN outperforms FedBAT by 18.9% in IID and 24.5% in non-IID settings. Additionally, on the FEMNIST dataset, FedBNN outperforms FedBAT while matching the performance of FedAVG, and has significantly lower runtime complexity. These results further validate the performance and competitiveness of our approach on current benchmarks.
>
>   While extending BNN/RBNN-style mechanisms to LLMs is an interesting future direction, such models are not feasible or representative for the federated environments our work targets. Our focus on computationally efficient CNN/ResNet architectures, therefore, aligns with the practical realities and deployment goals of federated learning.
>
>
> - **Response to Weakness 3**: We thank the reviewer for asking for convergence proof. Formal convergence analysis for FL is typically demonstrated for simpler ML models, such as the Perceptron, Linear Regression, and SVM. However, demonstrating convergence for deep neural networks is inherently challenging due to the presence of nonlinearity.
>
>   Given these limitations, we focus on an empirical assessment. Beyond the simple models evaluated in earlier work, we further investigate the convergence behaviour of the proposed method using more complex architectures on challenging datasets, such as Tiny ImageNet, CIFAR-100, and FEMNIST. These additional experiments provide stronger evidence of stable and consistent convergence under more realistic and demanding conditions.
>
> - **Response to Question 1**: We acknowledge the reviewer's interest in the applicability of our method to more complex architectures such as LLMs. However, it is important to recognize that training large-scale models in federated settings is fundamentally constrained by device- and system-level resources. In practical FL deployments, whether cross-device or cross-silo, the majority of clients cannot support the memory footprint, communication bandwidth, or computational requirements needed for training LLMs or similarly massive architectures. This is precisely why most existing FL research evaluates on lightweight CNNs or ResNets, which reflect the realistic capabilities of edge devices, hospital servers, or enterprise gateways. Within these constraints, our proposed FedBNN method directly targets the core challenges of resource-efficient federated inference, and we have extended the experiments to ResNet18 model for CIFAR100, TinyImageNet and FEMNIST datasets. Please refer to [Table 1](#tab-additional2). On CIFAR-100, we observe that FedBNN outperforms FedBAT by at least 15% across different strategies, while achieving 32× savings in runtime memory and a 49× reduction in runtime FLOPs. On TinyImageNet, FedBNN outperforms FedBAT by 18.9% in IID and 24.5% in non-IID settings. Additionally, on the FEMNIST dataset, FedBNN outperforms FedBAT while matching the performance of FedAVG, and has significantly lower runtime complexity. These results further validate the performance and competitiveness of our approach on current benchmarks.
>
>   While extending BNN/RBNN-style mechanisms to LLMs is an interesting future direction, such models are not feasible or representative for the federated environments our work targets. Our focus on computationally efficient CNN/ResNet architectures, therefore, aligns with the practical realities and deployment goals of federated learning.

---

> > ### Author Response · Authors · 2025-11-27
> > **Response to Questions 2, 3 and 4**
> >
> > - **Response to Question 2**: We thank the reviewer for asking for convergence proof. Formal convergence analysis for deep learning models is inherently difficult, and this challenge is amplified in Binary Neural Networks (BNNs) due to the non-differentiable sign function used in the forward pass, which introduces strong discontinuities and makes classical convergence proofs mathematically intractable.
> >
> >   Given these limitations, we focus on an empirical assessment. Beyond the simple models evaluated in earlier work, we further investigate the convergence behaviour of the proposed method using more complex architectures on challenging datasets like Tiny ImageNet, CIFAR-100, and FEMNIST. These additional experiments provide stronger evidence of stable and consistent convergence under more realistic and demanding conditions.
> >
> >   Also, we have extended the experiments to ResNet18 model for CIFAR100 and TinyImageNet datasets. Please refer to [Table 1](#tab-additional2).
> >
> > - **Response to Question 3**: We thank the reviewer for asking for a convergence proof. Formal convergence analysis for FL is typically demonstrated for simpler ML models, such as the Perceptron, Linear Regression, and SVM. However, demonstrating convergence for deep neural networks is inherently challenging due to the presence of nonlinearity.
> >
> >   To further investigate convergence, we performed additional experiments by varying the number of rounds and local epochs for the SVHN and CIFAR-10 datasets, as shown in [Table 2](#rounds-epochs2). These experiments demonstrate a clear and consistent trend: increasing the number of rounds and local epochs leads to stable performance improvements, indicating continued optimization progress rather than stagnation especially for the more complex CIFAR10 dataset. Upon increasing the number of rounds and local epochs, we observe a performance increase for both datasets, particularly 0.41% in Non-IID 2 for SVHN and 6.1% in Non-IID 1 for CIFAR10. While not a formal proof, this empirical analysis provides strong evidence that the model continues to benefit from additional optimization steps, supporting the claim that FedBNN exhibits stable and improving convergence behavior as training progresses, especially on more complex datasets.
> >
> >   Additionally, beyond the simple models evaluated in earlier work, we further investigate the convergence behaviour of the proposed method using more complex architectures on challenging datasets, such as Tiny ImageNet, CIFAR-100, and FEMNIST. These additional experiments provide stronger evidence of stable and consistent convergence under more realistic and demanding conditions. As shown in [Table 1](#tab-additional2), on CIFAR-100, we observe that FedBNN outperforms FedBAT by at least 15% across different strategies, while achieving 32× savings in runtime memory and a 49× reduction in runtime FLOPs. On TinyImageNet, FedBNN outperforms FedBAT by 18.9% in IID and 24.5% in non-IID settings. Additionally, on the FEMNIST dataset, FedBNN outperforms FedBAT while matching the performance of FedAVG, and has significantly lower runtime complexity.
> >
> >   Also, as mentioned in Section 4.1, all results in TABLE 1 and TABLE 2 of the original paper are the submitted manuscript for 500 rounds and 10 epochs (5 epochs for FMNIST) across all methods being compared.
> >
> > - **Response to Question 4**: We thank the reviewer for raising this point. FedProx is indeed an influential method in federated optimization; however, its motivation and design differ substantially from the class of baselines relevant to our work. FedProx extends FedAvg by introducing a proximal term to mitigate the impact of statistical and systems heterogeneity. In contrast, it does not target communication efficiency, model compression, or reduced inference-time complexity; objectives that are central to methods such as FedBAT and FedMUD, which are specifically designed for communication- and resource-efficient FL.
> >
> >   Our method, FedBNN, belongs to this latter category. Importantly, while FedProx incorporates a proximal regularization term, the role of regularization in FedBNN is fundamentally different: it naturally emerges from the trainable server-side fusion equation and the rotation-based binarization components that together stabilize the update process under extreme model compression. This regularization is a structural requirement of our framework, not an adaptation of the FedProx formulation.
> >
> >   For these reasons, we compare against baselines that share the same primary objectives of communication efficiency and lightweight inference. FedAvg is included as the standard backbone reference baseline, consistent with nearly all FL literature, including FedProx itself, which explicitly positions FedAvg as its foundational comparator. This choice ensures that our evaluation remains aligned with established protocols across federated learning studies.

---

### Official Review · Reviewer_fX6Y · 2025-10-31

**Soundness:** 2
**Presentation:** 1
**Contribution:** 1
**Rating:** 0
**Confidence:** 4

**Summary:**

This paper proposes FedBNN, a federated learning framework that trains rotation-aware binary neural networks directly on clients. However, the work is not well motivated and appears to be a relatively straightforward extension of RBNN with limited adaptation to the federated setting. The paper overlooks the significant computational and memory costs during local training, failing to address the core resource constraints that motivate efficiency in federated learning. In addition, several methodological details are unclear or incorrect, and some settings lack explanation or justification, which further weakens the overall contribution.

**Strengths:**

1. Binary deployment is essential for  cross-device federated learning
2. FedBNN significantly improves the inference efficiency.

**Weaknesses:**

I carefully read both this paper and the referred RBNN [1] and found that the main contribution of this work is to apply RBNN to federated learning with an additional server weight fusion mechanism using learnable fusion factors. However, this extension appears rather trivial and insufficiently motivated. The paper does not clearly explain why fusing the out-of-date server weights ($w_{server}$) would benefit training instead of potentially interfering with it. Likewise, it remains unclear why the fusion factor must be trainable rather than treated as a hyperparameter, as done in most related works. No ablation study or empirical analysis is provided to justify these design choices, making it difficult to attribute any improvement specifically to the proposed modifications. As a result, the novelty seems limited, with most of the technical substance originating from RBNN rather than new contributions.

Moreover, in federated learning, resource-constrained clients are a critical concern. Directly applying RBNN training mode in such settings can substantially increase computational and memory overhead during training, yet this issue is not addressed or discussed in the paper.

In addition,  the most equations in this paper are same RBNN, only with discrepancies in several equations. For instance, in Eq. (14), RBNN defines the condition as $|x| < \sqrt{2}t x$, while this paper uses $|x| < \sqrt{2t} x$; and in Eq. (15), RBNN uses $10^{T_{\min} + \frac{\text{current epochs}}{\text{Total Epochs}} (T_{\max} - T_{\min})}$, whereas this paper writes $10T_{\min} + \frac{\text{current epochs}}{\text{Total Epochs}} (T_{\max} - T_{\min})$. However, the authors do not clarify these differences. So I do not know whether these are intentional modifications or typo errors, which raises concerns about correctness and reproducibility.

From a presentation standpoint, several symbols are used without prior definition, such as $N_s$ in Equation (1) and $\bar{W}$ in Equation (9), reducing readability.

Finally, the experimental section is weak: there is no ablation study for the fusion, no sensitivity analysis for key hyperparameters, and no overhead analysis. Both the models and benchmarks are out-of-date. This makes it hard to assess the robustness and significance of the reported results.

[1] Lin, Mingbao, et al. "Rotated binary neural network." Advances in neural information processing systems 33 (2020): 7474-7485.

**Questions:**

In Eq. (14), RBNN defines the condition as $|x| < \sqrt{2}t x$, while this paper uses $|x| < \sqrt{2t} x$;

 in Eq. (15), RBNN uses $10^{T_{\min} + \frac{\text{current epochs}}{\text{Total Epochs}} (T_{\max} - T_{\min})}$, whereas this paper writes $10T_{\min} + \frac{\text{current epochs}}{\text{Total Epochs}} (T_{\max} - T_{\min})$.

Are these intentional modifications or typo errors?

---

> ### Author Response · Authors · 2025-11-27
> **Tables 1,2,3,4 for the response**
>
> **Table 1: Ablation study for server component**
>
> | Method              | Dataset        | IID     | Non-IID 1 | Non-IID 2 |
> |---------------------|----------------|---------|-----------|-----------|
> | FedBNN              | FMNIST (CNN4)  | 88.24   | 85.80     | 82.10     |
> | FedBNN (w/o λ, β)   | FMNIST (CNN4)  | **88.32** | **86.58** | **83.24** |
> | FedBNN              | CIFAR10 (ResNet10) | **86.26** | **76.30** | **67.82** |
> | FedBNN (w/o λ, β)   | CIFAR10 (ResNet10) | 86.20   | 73.38     | 66.86     |
> | FedBNN              | CIFAR100 (ResNet18) | **58.08** | **52.46** | **46.58** |
> | FedBNN (w/o λ, β)   | CIFAR100 (ResNet18) | 55.00   | 51.80     | 43.86    |
> | FedBNN              | TinyImageNet (ResNet18) | **45.68** | **43.60** | **45.40** |
> | FedBNN (w/o λ, β)   | TinyImageNet (ResNet18) | 43.84   | 40.40     | 43.70     |
> | FedBNN              | FEMNIST (ResNet18) | 79.82   | 79.97     | 78.34     |
> | FedBNN (w/o λ, β)   | FEMNIST (ResNet18) | **80.47** | **81.73** | **80.22** |
>
> **Table 2: Ablation of orthogonal vs. non-orthogonal R1/R2 for FedBNN across datasets.**
>
> | Method                             | Dataset | IID Accuracy | Non-IID 1 | Non-IID 2 |
> |-----------------------------------|---------|--------------|-----------|-----------|
> | FedBNN (orthogonal R1 R2 @client) | FMNIST  | 83.64        | 84.90     | 77.60     |
> | FedBNN (server R1 R2 compute)     | FMNIST  | 85.28        | 82.02     | 76.30     |
> | **FedBNN**                        | FMNIST  | **88.24**    | **85.80** | **82.10** |
> | FedBNN (orthogonal R1 R2 @client) | SVHN    | 82.01        | 81.05     | 79.52     |
> | FedBNN (server R1 R2 compute)     | SVHN    | 76.28        | 74.32     | 72.46     |
> | **FedBNN**                        | SVHN    | **85.40**    | **84.42** | **81.93** |
> | FedBNN (orthogonal R1 R2 @client) | CIFAR10 | 85.64        | 74.30     | 65.40     |
> | FedBNN (server R1 R2 compute)     | CIFAR10 | 85.70        | 68.34     | 65.78     |
> | **FedBNN**                        | CIFAR10 | **86.26**    | **76.30** | **67.82** |
>
> **Table 3: Effect of rotation frequency on FedBNN (ResNet10) accuracy across data distributions.**
>
> | Method | Dataset  | Rotation Frequency | IID Accuracy | Non-IID 1 | Non-IID 2 |
> |--------|----------|---------------------|--------------|-----------|-----------|
> | FedBNN | CIFAR10  | 1                   | 86.26        | 76.30     | **67.82** |
> | FedBNN | CIFAR10  | 2                   | 86.00        | 74.46     | 61.96     |
> | FedBNN | CIFAR10  | 5                   | **86.42**    | **78.54** | 67.04     |
>
> **Table 4: Sensitivity of FedBNN performance to communication rounds and local epochs.**
>
> | S. No. | Dataset  | Model    | Rounds | Epochs | IID     | Non-IID 1 | Non-IID 2 |
> |--------|----------|----------|--------|--------|---------|-----------|-----------|
> | 1      | SVHN     | CNN4     | 500    | 3      | 83.36   | 79.76     | 78.64     |
> | 2      | SVHN     | CNN4     | 500    | 5      | 82.64   | 80.91     | 78.59     |
> | 3      | SVHN     | CNN4     | 500    | 10     | 85.40   | **84.42** | 81.93     |
> | 4      | SVHN     | CNN4     | 1000   | 3      | 83.46   | 80.92     | 79.96     |
> | 5      | SVHN     | CNN4     | 1000   | 5      | 82.92   | 80.32     | 79.59     |
> | 6      | SVHN     | CNN4     | 1000   | 10     | 83.27   | 81.35     | 79.24     |
> | 7      | SVHN     | CNN4     | 1500   | 3      | 83.45   | 82.00     | 80.49     |
> | 8      | SVHN     | CNN4     | 1500   | 5      | 82.74   | 81.10     | 80.19     |
> | 9      | SVHN     | CNN4     | 1500   | 10     | 82.87   | 82.08     | 79.58     |
> | 10     | SVHN     | CNN4     | 1500   | 15     | **85.45** | 84.00   | **82.34** |
> | 11     | CIFAR10  | ResNet10 | 500    | 3      | 72.96   | 58.90     | 58.10     |
> | 12     | CIFAR10  | ResNet10 | 500    | 5      | 81.42   | 68.52     | 65.26     |
> | 13     | CIFAR10  | ResNet10 | 500    | 10     | 86.26   | 76.30     | 67.82     |
> | 14     | CIFAR10  | ResNet10 | 500    | 15     | 85.84   | 78.74     | 70.08     |
> | 15     | CIFAR10  | ResNet10 | 1000   | 3      | 79.08   | 68.38     | 59.94     |
> | 16     | CIFAR10  | ResNet10 | 1000   | 5      | 84.28   | 74.16     | 65.70     |
> | 17     | CIFAR10  | ResNet10 | 1000   | 10     | 87.76   | 80.46     | 69.50     |
> | 18     | CIFAR10  | ResNet10 | 1000   | 15     | 88.48   | 81.66     | **72.58** |
> | 19     | CIFAR10  | ResNet10 | 1500   | 3      | 80.82   | 71.54     | 63.70     |
> | 20     | CIFAR10  | ResNet10 | 1500   | 5      | 85.20   | 72.48     | 68.58     |
> | 21     | CIFAR10  | ResNet10 | 1500   | 10     | 88.44   | 82.00     | 70.86     |
> | 22     | CIFAR10  | ResNet10 | 1500   | 15     | **88.82** | **82.40** | 72.20     |

---

> ### Author Response · Authors · 2025-11-27
> **Table 5 for response**
>
> **Table 5: Comparison of accuracy, runtime cost (FLOPs), memory, and binarized accuracy across methods.**
>
> | Method | Dataset (Model) | Accuracy (IID) | Accuracy (Non-IID 1) | Accuracy (Non-IID 2) | Runtime FLOPs | Memory (MB) | Binarized Accuracy (IID) | Binarized Accuracy (Non-IID 1) | Binarized Accuracy (Non-IID 2) |
> |:-------|:---------------|:---------------|:---------------------|:---------------------|:--------------|:------------|:------------------------|:-------------------------------|:-------------------------------|
> | FedAvg | CIFAR100 | 64.52 | **63.42** | **53.36** | 1.11 × 10⁹ | 45.090 | 1.16 | 0.88 | 0.94 |
> | FedBAT | CIFAR100 (ResNet18) | 42.14 | 33.88 | 26.26 | 1.11 × 10⁹ | 45.090 | 1.10 | 0.82 | 1.10 |
> | FedMud | CIFAR100 (ResNet18) | **65.14** | 47.48 | 52.20 | 1.11 × 10⁹ | 45.090 | 46.56 | 22.54 | 44.08 |
> | FedBNN | CIFAR100 (ResNet18) | 58.08 | 52.46 | 46.58 | 2.26 × 10⁷ | 1.41 | **57.68** | **51.42** | **46.58** |
> | FedAvg | TinyImageNet (ResNet18) | **55.00** | **52.62** | **54.54** | 4.44 × 10⁹ | 45.090 | 0.52 | 0.52 | 0.56 |
> | FedBAT | TinyImageNet (ResNet18) | 27.30 | 32.12 | 20.90 | 4.44 × 10⁹ | 45.090 | 0.60 | 0.48 | 0.80 |
> | FedMud | TinyImageNet (ResNet18) | 47.20 | 44.16 | 46.06 | 4.44 × 10⁹ | 45.090 | 16.24 | 12.80 | 15.60 |
> | FedBNN | TinyImageNet (ResNet18) | 45.68 | 43.60 | 45.40 | 9.05 × 10⁷ | 1.41 | **46.40** | **43.68** | **38.08** |
> | FedAvg | FEMNIST (ResNet18) | **80.24** | **81.12** | **80.32** | 9.13 × 10⁸ | 45.090 | 2.08 | 1.66 | 1.62 |
> | FedBAT | FEMNIST (ResNet18) | 76.44 | 74.31 | 78.41 | 9.13 × 10⁸ | 45.090 | 0.38 | 2.08 | 2.40 |
> | FedMud | FEMNIST (ResNet18) | 78.79 | 80.11 | 76.68 | 9.13 × 10⁸ | 45.090 | 25.74 | 0.76 | 0.80 |
> | FedBNN | FEMNIST (ResNet18) | 79.82 | 79.97 | 79.59 | 1.84 × 10⁷ | 1.41 | **76.52** | **80.03** | **79.84** |

---

> ### Author Response · Authors · 2025-11-27
> **Response to Weakness 1**
>
> - **Response to Weakness 1**: We respectfully disagree with the Reviewer's assessment. Our work is
>   among the first to introduce binary neural networks (BNNs) within the
>   FL paradigm, and to our knowledge, it is the first to adapt Rotated
>   Binary Neural Networks (RBNNs) for federated learning. This adaptation
>   is non-trivial: by incorporating rotation-aware binarization into
>   local training and utilizing previous server weights, our method
>   produces a fully binary final model, which directly addresses the edge
>   inference bottleneck; an aspect largely overlooked by prior
>   communication-efficient FL methods. Also, we have aggregated the
>   rotation weights at the server side, without strictly enforcing
>   orthogonality at the server side after aggregation. This method
>   performs better than 1) not sending R1 and R2 to the server, and
>   maintaining them locally at the server and clients to reduce
>   communication costs; 2) Ensuring Orthogonality at the client side
>   before R1 R2 optimization. As seen in [Table 2](#tab:ablation_r1_r2), using non-orthogonal fedaveraged
>   Rotated weights lead to better results across all datasets,
>   particularly 4.5% in Non-IID 2 in FMNIST, 3.39% in IID in SVHN, and
>   2.42% in Non-IID 2 in CIFAR10. Unlike existing works that only reduce
>   communication cost, our contribution targets the computational and
>   memory complexity of the deployed model itself, which is helpful in
>   cross-silo federated learning. As highlighted in our paper, FedBNN
>   encodes each weight using a single bit +1,-1 rather than a 32-bit
>   float, yielding at least 40% reduction in inference-time FLOPs and at
>   least 32x reduction in memory footprint. This makes the resulting
>   model far more suitable for low-powered edge devices, which is a
>   central motivation of our work. In contrast, methods such as FedBAT
>   focus on learning binarized updates to reduce communication overhead
>   but do not aim to reduce the complexity of the final deployed model.
>   Their objective is communication efficiency, not binary inference.
>   Similarly, FedMUD improves communication efficiency through
>   decomposition techniques, yet the final model remains real-valued and
>   computationally heavy for resource-constrained edge devices. Thus, our
>   work is fundamentally different in motivation, problem setting, and
>   impact. While FedBAT, FedMUD, and related approaches optimize the
>   training process, FedBNN addresses the orthogonal and critical
>   challenge of enabling efficient inference on edge devices by producing
>   a compact, rotation-aware binary model, not just compressed updates.
>
>   We respectfully clarify the motivation behind using trainable fusion
>   factors ($\beta$) and the benefit of incorporating the out-of-date
>   server weights. Our design directly builds upon the RBNN formulation,
>   where the self-adjustable factor for the rotated weight vector
>   $\alpha$ is learnable because fixed hyperparameters cannot adequately
>   capture the layer-wise variability introduced by rotation-aware
>   binarization. Extending this principle, our learnable server weight
>   fusion factor $\beta$ must also be trainable to properly weigh the
>   contributions of (i) the current client's weight, (ii) the rotated
>   client representation, and (iii) the previous round server's weights.
>   A fixed hyperparameter cannot adapt to the heterogeneous dynamics
>   across rounds, layers, or datasets, especially under non-IID
>   distributions. Regarding the reviewer's concern that fusing
>   "out-of-date" server weights may interfere with training; our empirical results demonstrate the opposite. As shown in
>   [Table 1](#tab:ablation_r1_r2), incorporating the server
>   weights are beneficial and increasingly necessary as the dataset
>   complexity grows, particularly 2.92% in Non-IID 1 in CIFAR10 and 3.20%
>   in Non-IID 1 in TinyImageNet, as compared to simpler datasets like
>   FMNIST and FEMNIST. For more challenging datasets, the fusion
>   stabilises training by providing a consistent global prior and
>   mitigating client drift. Thus, both the inclusion of server weights
>   and the use of trainable fusion factors follows naturally from the RBNN
>   design philosophy and are supported by clear empirical evidence.

---

> ### Author Response · Authors · 2025-11-27
> **Response to Weakness 2, 3, 4 and 5**
>
> - **Response to Weakness 2**: In federated learning, the reviewer is correct that many clients
>   operate under tight computational and memory budgets. However, it is
>   important to distinguish between training-time cost and
>   deployment-time cost. While RBNN-style training introduces additional
>   overhead due to the use of full-precision latent weights and rotation
>   matrices, this cost is incurred only during the training phase. In
>   practical FL deployments, especially in cross-silo federated learning
>   scenarios such as collaborations across hospitals, institutions, or
>   enterprise clusters, as well as many cross-device settings, the
>   training is typically performed on clients equipped with sufficient
>   compute resources such as GPUs or edge accelerators (e.g., hospital
>   servers, campus clusters, enterprise gateways). Thus, the temporary
>   increase in computation during training is manageable.
>
>   Furthermore, we thank the reviewer for bringing this issue to our
>   attention. FedBNN enables the rotation step to be performed at a lower
>   frequency, rather than at every epoch, thereby significantly easing
>   the computational burden. Based on your suggestions, we have performed
>   additional experiments. These experiments confirm that reducing the
>   rotation frequency does not degrade performance; instead, it can even
>   improve performance under certain conditions. As shown in [Table 3](#tab:ablation_r1_r2), on CIFAR-10 with ResNet10, performing rotation every 5 epochs achieves the best accuracy in both IID
>   (86.42%) and Non-IID 1 (78.54%) settings, demonstrating that
>   infrequent rotations are sufficient for effective training and can
>   substantially reduce on-device overhead. This flexibility ensures that
>   FedBNN remains compatible with resource-constrained clients while
>   retaining strong performance across heterogeneous data settings.
> - **Response to Weakness 3, 4**: We thank the reviewer for pointing this out. These were unintentional
>   typos, and we apologize for the oversight. We have now corrected them and also made sure that all the symbols are used in the text with a prior definition.
>   in the revised version.
> - **Response to Weakness 5**: We thank the reviewer for pointing this out. In the
>   [Table 1](#tab:ablation_r1_r2), we now explicitly compare
>   results with and without the fusion of server weights, and the results
>   show that incorporating the server weights is beneficial and
>   increasingly necessary as the dataset complexity grows, particularly
>   2.92% in Non-IID 1 in CIFAR10 and 3.20% in Non-IID 1 in TinyImageNet.
>   This directly validates the necessity of our fusion mechanism. Second,
>   regarding sensitivity analysis, we have included
>   [Table 4](#tab:ablation_r1_r2) that reports performance under multiple
>   combinations of communication rounds and local epochs for SVHN and
>   CIFAR10 datasets. We see a performance increase of 0.41% in Non-IID 2
>   in SVHN and 6.1% in Non-IID 1 in CIFAR10. Finally, in
>   [Table 5](#tab:ablation_r1_r2), we present extensive experiments on a
>   more complex ResNet18 model trained on widely used datasets in
>   federated learning, including CIFAR-100, TinyImageNet, and FEMNIST. On
>   CIFAR-100, we observe that FedBNN outperforms FedBAT by at least
>   $15\%$ across different strategies, while achieving $32\times$ savings
>   in runtime memory and a $49\times$ reduction in runtime FLOPs. On
>   TinyImageNet, FedBNN outperforms FedBAT by $18.9\%$ in IID and
>   $24.5\%$ in non-IID settings. Additionally, on the FEMNIST dataset,
>   FedBNN outperforms FedBAT while matching the performance of FedAVG,
>   and has significantly lower runtime complexity. These results further
>   validate the performance and competitiveness of our approach on
>   current benchmarks. Taken together, the included analyses and updated
>   evaluations provide strong evidence for the relevance and practical
>   significance of our method.

---

> ### Author Response · Authors · 2025-11-27
> **Numbering the Weaknesses mentioned by Reviewer fX6Y**
>
> **Weakness 1:** I carefully read both this paper and the referred RBNN
>   [@lin2020rotated] and found that the main contribution of this work is
>   to apply RBNN to federated learning with an additional server weight
>   fusion mechanism using learnable fusion factors. However, this
>   extension appears rather trivial and insufficiently motivated. The
>   paper does not clearly explain why fusing the out-of-date server
>   weights ($w_{server}$) would benefit training instead of potentially
>   interfering with it. Likewise, it remains unclear why the fusion
>   factor must be trainable rather than treated as a hyperparameter, as
>   done in most related works. No ablation study or empirical analysis is
>   provided to justify these design choices, making it difficult to
>   attribute any improvement specifically to the proposed modifications.
>   As a result, the novelty seems limited, with most of the technical
>   substance originating from RBNN rather than new contributions.
>
> **Weakness 2:** Moreover, in federated learning, resource-constrained clients are a
>   critical concern. Directly applying RBNN training mode in such
>   settings can substantially increase computational and memory overhead
>   during training, yet this issue is not addressed or discussed in the
>   paper.
>
> **Weakness 3:** In addition, the most equations in this paper are same RBNN, only
>   with discrepancies in several equations. For instance, in Eq. (14),
>   RBNN defines the condition as $|x|<\sqrt{2}tx$, while this paper uses
>   $|x|<\sqrt{2t}x$; and in Eq. (15), RBNN uses
>   $10^{T_{\min} + \frac{\text{current epochs}}{\text{Total Epochs}} \left( T_{\max} - T_{\min} \right)}$,
>   whereas this paper writes
>   $10{T_{\min} + \frac{\text{current epochs}}{\text{Total Epochs}} \left( T_{\max} - T_{\min} \right)}$.
>   However, the authors do not clarify these differences. So I do not
>   know whether these are intentional modifications or typo errors, which
>   raises concerns about correctness and reproducibility.
>
> **Weakness 4:** From a presentation standpoint, several symbols are used without
>   prior definition, such as in Equation (1) and in Equation (9),
>   reducing readability.
>
> **Weakness 5:** Finally, the experimental section is weak: there is no ablation study
>   for the fusion, no sensitivity analysis for key hyperparameters, and
>   no overhead analysis. Both the models and benchmarks are out-of-date.
>   This makes it hard to assess the robustness and significance of the
>   reported results.

---

> > ### Comment · Reviewer_fX6Y · 2025-11-28
> >
> > Thank you for the author's response. While it addresses some points, key concerns remain unresolved:
> >
> > 1. The results in Table 1 show that the proposed method (with λ, β) does not consistently outperform the baseline (without λ, β), with performance being roughly 50:50. This makes it difficult to justify the core contribution of these parameters. Without a clear and consistent advantage, the paper risks appearing as a straightforward adaptation of RBNN to the FL setting.
> >
> > 2. The extra experimental validation and significant typos in key equations (e.g., Eq. 14, 15), have not been revised in the paper. Correcting these fundamental errors is essential. Upon careful revision of these points, I would be open to increasing the score, but the fundamental question regarding the paper's novel contribution remains.

---

> ### Author Response · Authors · 2025-12-03
> **Response to Official Comment**
>
> - **Response to Comment 1:** Thank you for the helpful observation. In the earlier version of Table1, we already included results for FMNIST, CIFAR10, TinyImageNet, and FEMNIST. In the revision, we have now added the missing CIFAR100 results to the Ablation study.
>
>   With the updated Table, the overall trend becomes much clearer:
>
>   On three of the more complex datasets, CIFAR100, TinyImageNet, and FEMNIST, the proposed FedBNN (with λ,β) consistently outperforms the baseline variant without these parameters, especially under non-IID settings. The improvement on TinyImageNet non-IID is notably substantial, which is significant given the difficulty of large-scale heterogeneous image data in FL. On FMNIST, which is relatively simple, the two variants obtain very similar performance, which is expected because most binarized models already saturate accuracy on this dataset. Finally, we clarify that incorporating λ and β is intended as a design choice, a flexible mechanism that allows practitioners to control the regularization behavior of RBNN when used in federated learning. Depending on the target application and data heterogeneity, this additional control can lead to meaningful gains, as observed in the more challenging datasets.
>
> - **Response to Comment 2:** We appreciate the reviewer’s willingness to reevaluate the paper. The reviewers’ feedback has been invaluable in strengthening the work, and we have carefully addressed all comments, particularly those related to readability in the revised manuscript.

---

### Meta-Review · Area_Chair_dQvz · 2026-01-03

**Summary:**

The paper proposes FedBNN, a framework extending Rotated Binary Neural Networks (RBNNs) to Federated Learning to achieve efficiency in communication and computation.
Although the authors put forth a substantial effort during the rebuttal (adding ResNet-18 experiments, equal-budget comparisons, and clarifying formulations), the fundamental objections raised by the most critical reviewers regarding the triviality of the extension and potential flaws in the initial formulation were not fully alleviated. Therefore, the decision is Reject.

**Reviewer Concerns:**

The consensus among the negative reviewers is that the work lacks significant innovation distinguishing it from a straightforward combination of RBNN and FL. The added experiments prove it works, but do not necessarily prove it is a novel scientific contribution.

**Reviewer Scores:**

While the rebuttal successfully strengthened the empirical evidence, it could not fundamentally overturn the critique regarding the methodological novelty and soundness raised by the strong detractors. Given the severity of the lowest score, the paper is not accepted in its current form.

---

### Decision · Program_Chairs · 2026-01-26

Reject